# `SEM-CTRL`: Semantically Controlled Decoding

**Mohammad Albinhassan**                                       *m.albinhassan23@imperial.ac.uk*
*Department of Computing*
*Imperial College London*

**Pranava Madhyastha**                                       *pranava.madhyastha@city.ac.uk*
*Department of Computer Science*
*City, University of London*

**Alessandra Russo**                                       *a.russo@imperial.ac.uk*
*Department of Computing*
*Imperial College London*

**Reviewed on OpenReview:** *https://openreview.net/forum?id=ICUHKhOISN*

## Abstract

Ensuring both syntactic and semantic correctness in Large Language Model (LLM) outputs remains a significant challenge, despite being critical for real-world deployment. In this paper, we introduce `SEM-CTRL`, a unified approach that allows for enforcing rich context-sensitive constraints, and task and instance specific semantics directly on the LLM decoder. Our approach integrates token-level MCTS which is guided by specific syntactic and semantic constraints. The constraints over desired outputs are expressed using Answer Set Grammars, which is a logic-based formalism that generalizes context sensitive grammars while incorporating background knowledge to represent task-specific semantics. We show that our approach helps guarantee valid completions for any off-the-shelf LLM without the need for fine-tuning. We evaluate `SEM-CTRL` on a range of tasks, including synthetic grammar synthesis, combinatorial reasoning, JSON parsing, and planning. Our experimental results demonstrate that `SEM-CTRL` allows even small pre-trained LLMs to efficiently outperform larger variants and state-of-the-art reasoning models (e.g., *o4-mini*) while simultaneously guaranteeing semantic validity.

## 1 Introduction

Controlled generation aims at designing better decoding methodologies for Large Language Models (LLMs), which guarantees output validity according to formal specifications (Welleck et al., 2024). The challenge of controlled generation has so far seen many appealing approaches that can be broadly categorized as: (i) syntactic control – such as constraining based on regular or Context-Free Grammars (CFGs) (Geng et al., 2023; Beurer-Kellner et al., 2024, inter alia); (ii) control based on domain-specific semantic constraints (e.g., Scholak et al., 2021; Poesia et al., 2022); and (iii) search-guided reasoning – improving control (e.g., Zhang et al., 2023b; Hao et al., 2023).

Nevertheless, the effectiveness of existing approaches remains limited. Syntactic control is insufficient for real-world tasks demanding context-sensitive correctness based on a token's relative position in a sequence (Scholak et al., 2021). Domain-specific solutions lack generalizability across tasks (Poesia et al., 2022; Roy et al., 2023). Critically, both focus exclusively on ensuring the LLM's generations conform to some specification (validity) without explicitly encoding the notion of solving the task correctly (correctness). Search-based methods, while designed to capture correctness, suffer from inefficient exploration and premature pruning of valid solutions precisely because they do not explicitly capture validity (e.g., Zhang et al., 2023b; Wan et al., 2024). These empirical findings suggest fundamental limitations in current frameworks for handling both syntactic and semantic constraints simultaneously, as well as for expressing correctness.

We conjecture that the most plausible reason for these limitations is that existing approaches either focus solely on local constraints or lack semantic guidance during search. In this paper, we introduce a new approach, called *Semantically Controlled Decoding* (`SEM-CTRL`), that unifies semantic constraints with guided search to enable robust, controlled generation. `SEM-CTRL` exploits Answer Set Grammars (ASGs) to specify both syntactic structures and semantic rules within a single formalism, and combines this with token-level Monte-Carlo Tree-Search (MCTS), guided by domain-specific rewards. `SEM-CTRL` distinguishes itself from prior work through: (1) directly encoding Context-Sensitive Grammars (CSGs) to capture validity, versus simpler CFGs (e.g., Geng et al., 2023) or ad-hoc constraint checks (e.g., Scholak et al., 2021); (2) using ASGs to capture semantic meanings; and (3) employing MCTS for global correctness optimization.

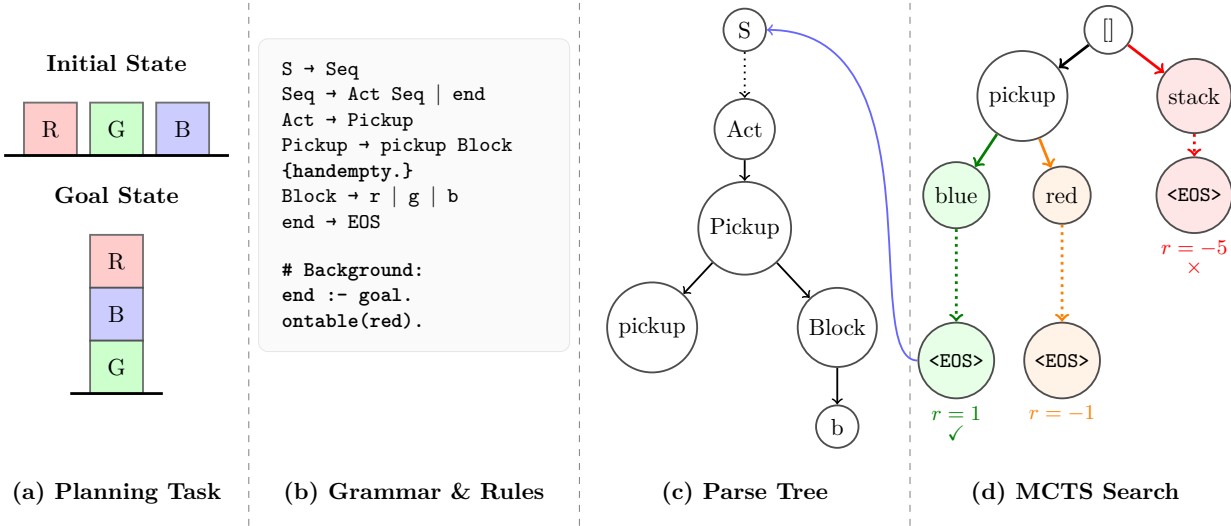

Figure 1: Overview of `SEM-CTRL` showing: (a) Blocksworld planning task with initial and goal states. (b) ASG fragment showing syntax and semantic rules. Curly braces {...} denote parse tree semantic constraints $\Psi_{PR}$, with domain rules and state encoding $\Psi_B$ under "Background". (c) Partial parse tree of a valid solution sequence. (d) MCTS search over the token space, with correct (green / ✓), suboptimal (orange / $-2$), and invalid paths (red / ×). States are simplified to show the generated token at each node instead of the entire sequence. Blue arrow shows MCTS-parse tree correspondence; each node maps to a valid parse tree as constraints are computed per step.

As an illustration, consider the Blocksworld planning task presented in Figure 1 (a), where an agent must rearrange a series of colored boxes into a target configuration. We distinguish between two critical dimensions: *validity* and *correctness*. Validity requires satisfying formal constraints. Traditional syntactic controls (e.g., CFGs) fail here as they cannot express context-dependent rules—a sequence may be syntactically valid yet selects invalid actions given the current state. ASGs ensure validity by enforcing semantic preconditions (e.g., one cannot pick up a block if the hand is full). Correctness, however, requires solving the downstream task. A semantically valid sequence may be functionally inadequate, such as repeatedly manipulating blocks without progressing towards the goal state (e.g., `pickup(A), putdown(A)` is valid but not correct if the goal is to stack `A` on `B`). Global correctness optimization refers to inference-time methods whose optimization objective is defined over entire sequences, evaluating decisions based on their contribution to the final task outcome (e.g., `pickup(A), stack(A,B)`). This contrasts with local optimization approaches, which optimize step-wise objectives such as next-token likelihood or locally constrained probabilities (e.g., greedy or locally constrained decoding), without explicitly optimizing for task success over full sequences (Geng et al., 2023; Zhang et al., 2023b; Welleck et al., 2024, inter alia). Unconstrained search methods attempt this optimization but must prune the token action-space due to computational feasibility, risking the elimination of the solution and thereby losing reachability guarantees. `SEM-CTRL` unifies these by performing global correctness optimization strictly within the guaranteed semantically valid space defined by the ASG (Figure 1 (b) and (d)).

Our empirical results show that `SEM-CTRL` successfully addresses these challenges through three key contributions: 1) a domain-independent framework using ASGs to capture a comprehensive hierarchy of token-aligned constraints; 2) an efficient token-level MCTS procedure that explores only semantically valid trajectories; and finally 3) using experiments across Synthetic Grammar Synthesis, Combinatorial Reasoning, JSON parsing, and Planning, we demonstrate that `SEM-CTRL` enables even smaller language models (such as with 1B parameters), to outperform larger state-of-the-art reasoning specific models (such as o1-preview, o4-mini, and DeepSeek-R1), while guaranteeing output correctness. The results underscore the importance of jointly capturing validity and correctness, which is underexplored in the current research landscape.

## 2 Background

In this section, we present the relevant background for `SEM-CTRL`, and introduce formal languages and ASGs.

**Formal Languages and Grammars** A formal language $L$ is a (possibly infinite) set of strings composed of a finite alphabet or vocabulary $\Sigma$, adhering to specific rules governing syntax and semantics. Each *word* in the language, $w = s_0, \cdots, s_i$, is a finite sequence of *symbols* $s_k \in \Sigma$. The set of all possible strings (including the empty string $\epsilon$) over $\Sigma$ forms the Kleene closure $\Sigma^*$, with $L \subseteq \Sigma^*$. Languages are generated by means of formal grammars $G = \langle N, T, P, S \rangle$, where $N$ is a finite set of non-terminal symbols, $T$ is the set of terminal symbols ($T = \Sigma$), $P$ is a finite set of production rules of the form $A \rightarrow \alpha$ (where, the left hand side $A \in N$ and the right hand side $\alpha \in (N \cup T)^*$), and $S \in N$ is the start symbol. A string $w$ belongs to the language of a grammar $G$, denoted by $w \in L(G)$, if there exists a sequence of rule applications (derivation) from $S$ to $w$. A parse tree $PT$ then represents this derivation hierarchically, with internal nodes labeled by $N$, leaves by $T$, and edges showing the application of production rules (Linz & Rodger, 2022).

The expressiveness of formal languages varies with grammar restrictions. CFGs restrict productions to $A \rightarrow \alpha$ where $A \in N$, thus suitable for nested structures like natural language syntax. Context-Sensitive Grammars (CSGs) allow more expressive productions of the form $\alpha A \beta \rightarrow \alpha \gamma \beta$ where $\alpha, \beta \in (N \cup T)^{*1}$, and $\gamma \in (N \cup T)^{+2}$. This enables CSGs to capture context-dependent patterns – e.g., while CFGs can generate $L_1 = \{a^i b^j c^k \mid i, j, k \geq 0\}$, only CSGs can generate $L_2 = \{a^n b^n c^n \mid n \geq 0\}$ where counts are equal.

**Answer Set Grammars** ASG is a symbolic framework for expressing CSGs that extend CFG production rules with context-sensitive constraints (for a thorough introduction, we refer the reader to Law et al., 2019). An ASG is composed of a CFG, augumented with a set $\Psi_{PR}$ of context-sensitive constraints annotating the CFG's productions rules, and a domain knowledge $\Psi_B$, capturing domain-specific semantic information as general rules and instance-specific facts. Both context-sensitive constraints and domain knowledge are expressed using Answer Set Programming (ASP), a symbolic formalism for representing knowledge and performing reasoning using a specialized symbolic solver. In ASP, general rules are of the form $h :- b_1, \cdots, b_n$ (read as "$h$ is true if all $b_i$ are true") and constraints are of the form $:- b_1, \cdots, b_n$ (ruling out solutions where all $b_i$'s are satisfied). A string $w$ is said to belong to the language of a given ASG (i.e., $w \in L(G_{ASG})$) if there exists a parse tree $PT$ from $S$ to $w$ that satisfies all the constraints and domain knowledge specified in the given ASG. This is formally defined by Equation (1), where $\text{str}(PT)$ returns the string generated by the parse tree $PT$, and $sat(G[PT])$ defines the satisfiability (based on the ASP semantics (Lifschitz, 2019)) of the ASP program $G[PT]$ that encodes the parse tree $PT$ and $\Psi = \Psi_{PR} \cup \Psi_B$.

$$w \in L(G_{ASG}) \iff \exists PT : \text{str}(PT) = w \land \text{sat}\big(G[PT]\big), \tag{1}$$

In other words, the set of logical statements, rules, and constraints encoded in $\Psi$ must all be true simultaneously in the parse tree of $w$. ASGs inherit the expressiveness of ASP, which is a state-of-the-art computational first-order logic paradigm capable of capturing first-order logic with default and preference reasoning, and of efficiently solving combinatorial search problems (Lifschitz, 2019; Erdem et al., 2016). While ASGs do not natively support certain logics such as fuzzy logic, alternative solvers could extend this capability, potentially extending into other NLP applications. For the class of grammars used in this paper, constraint checking

---

[1]zero or more elements from the set
[2]one or more elements from the set

has exponential worst-case complexity; we refer readers to Law et al. (2019) for detailed complexity analysis and proofs, and to Section 5.4 for empirical performance across our benchmark tasks.

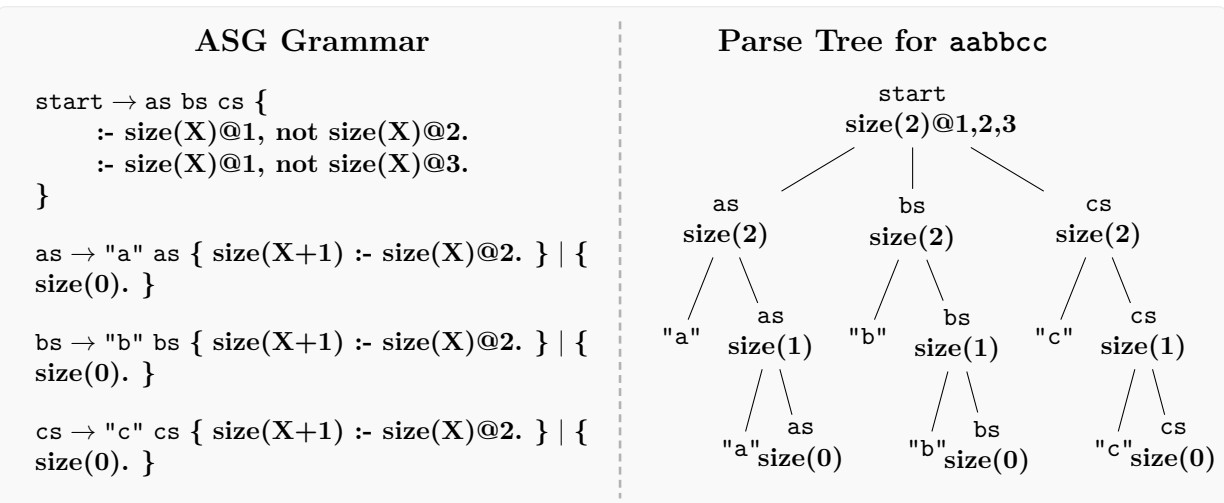

Figure 2: ASG for $a^n b^n c^n$ (left) and corresponding parse tree for `aabbcc` with ASP annotations (right). The grammar uses ASP annotations to enforce equal sequence lengths, with nodes showing computed size.

Figure 2 illustrates the ASG for the above mentioned language $L_2 = \{a^n b^n c^n \mid n \geq 0\}$ alongside its corresponding parse tree for the string `aabbcc`. The constraints in curly brackets $\{\dots\}$ (shown in bold) represent the parse tree annotations $\Psi_{PR}$ expressed as ASP code. $\Psi_B$ is, in this case, empty. The predicate `size(X)` tracks the length $n$ of each sequence, where `@i` indicates the $i$-th child position in the production rule. The constraint `:- size(X)@1, not size(X)@2.` reads as: "if the first child has size $X$, but the second child does not have size $X$, then reject this parse". The production rules (except for the first one) contain two alternatives separated by '|': the first alternative outputs one terminal symbol (i.e., a) and increments the size counter by one, using the value propagated from the recursively generated second child (i.e., `size(X)@2`), via the rule `size(X+1) :- size(X)@2`. The second branch initializes the base case counter with `size(0)`. The parse tree demonstrates how these annotations are computed: each non-terminal node shows its computed size value, building up from `size(0)` to `size(2)` at the sequence roots. The parse tree root node accepts the string `aabbcc` if all three child sequences have equal length. This is verified by checking the satisfiability of the two constraints. Note that, in Figure 2, if all annotations in the curly brackets were empty (i.e., $\Psi = \emptyset$), the corresponding ASG would reduce to the CFG language $L_1 = \{a^i b^j c^k \mid i, j, k \geq 0\}$.

Authoring new ASGs requires both domain knowledge of the underlying task and technical expertise in ASG construction. The latter consists of two parts: defining the CFG and specifying semantic constraints (see Appendix D for additional details and ASG examples that provide concrete illustrations of this separation). CFGs are widely adopted in practice, with existing grammars and tooling available for many tasks (e.g., our JSON grammar in Section 5.5 was automatically generated from the Lark Python parsing library). Semantic constraints require expertise in ASP, though ASP has proven to be effective across many application domains including knowledge representation, robotics, and bioinformatics (for comprehensive examples, see Erdem et al., 2016). While this expertise requirement may pose an adoption barrier, recent work demonstrates that LLMs can generate ASP code from natural language specifications (e.g., Alviano et al., 2025; Schrader et al., 2025; Ishay et al., 2023), potentially lowering this barrier for practitioners in less formalized domains.

## 3 Our Approach: SEM-CTRL

We now present `SEM-CTRL`, which, at its core, uses ASGs to enable token-level constraint verification during generation, guided by semantically controlled MCTS. This approach ensures both semantic validity and solution quality: the constraints guarantee validity (by construction), while tree search efficiently explores the semantically valid token space to find correct solutions.

### 3.1 Controlled Decoding through Semantic Constraints

To achieve controlled and valid generation, we require that every partial output of the language model remains extendable to a complete sequence in a predetermined target language. Equivalently, at each generation step we restrict admissible tokens to preserve the possibility of producing a valid final string. We achieve this by defining a general constraint function that maps partial sequences to valid next tokens, then instantiate this framework with constraints of increasing expressivity.

Concretely, we now formalize *semantic control* in autoregressive generation. Let $\mathcal{V}$ be the vocabulary of the LLM defined by its tokenizer and $\mathcal{V}^*$ its Kleene closure. We aim to ensure that any generated sequence belongs to a target formal language $L \subseteq \mathcal{V}^*$. We define a *constraint function* $\mathcal{C} : \mathcal{V}^* \to \mathcal{V}$ that maps any prefix $y_{<t} = (y_1, \cdots, y_{t-1}) \in \mathcal{V}^*$ to its valid next tokens:

$$\mathcal{C}(y_{<t}) = \left\{ y_t \mid \exists\, w \in L : (y_{<t} \circ y_t)\, \text{is a prefix of}\, w \right\}, \tag{2}$$

where $\circ$ denotes token concatenation. Intuitively, the function $\mathcal{C}(y_{<t})$ returns the set of possible derivation steps from $y_{<t}$ (set of tokens) that may yield a valid string $w \in L$.

Given a language $L$, Equation (2) defines how a set of valid tokens can be generated. Depending on $L$, $\mathcal{C}(y_{<t})$ encodes different levels of control.

**Definition 3.1** (Syntactic Control). Given a CFG $G$, and its associated language $L(G)$, $\mathcal{C}_{\text{CFG}}$ is the constraint function that returns tokens governed by the CFG derivation steps.

**Definition 3.2** (Context-Sensitive Control). Given a CSG $G$, and its associated language $L(G)$, $\mathcal{C}_{\text{CSG}}$ is the constraint function that returns tokens governed by the CSG derivation steps, i.e., that satisfy the context-sensitive constraints in $\Psi_{PR}$ with $\Psi_B = \emptyset$.

**Definition 3.3** (Semantic Control). Given an ASG $G$ with $\Psi_B \neq \emptyset$, and its associated language $L(G)$, $\mathcal{C}_{\text{SEM}}$ is the constraint function that returns tokens governed by the ASG derivation steps, i.e., that satisfy the semantic constraints in $\Psi_{PR}$ and $\Psi_B$.

As an illustration, consider automated planning domains. $\mathcal{C}_{\text{CFG}}$ ensures parseable action sequences. $\mathcal{C}_{\text{CSG}}$ enforces type consistency of action predicates. $\mathcal{C}_{\text{SEM}}$ guarantees that the action sequence in $y_{<t}$ maps to valid, executable operations in the target environment — maintaining state consistency and avoiding invalid trajectories. We note that the semantic constraints capture domain knowledge about what makes action sequences meaningful in the real planning environment. This is because semantic constraints refer to domain-specific knowledge, meaningful relationships, and rules that transcend typical local, positional nature of (CFG or CSG) grammar rules.

**Controlled Autoregressive Sampling** To incorporate these constraint functions at inference, we define a constrained sampling mechanism. Let $x$ denote the input and $y_{<t}$ the prefix of tokens generated up to timestep $t-1$. The language model defines the autoregressive conditional distribution $p_\theta(y_t \mid x, y_{<t})$. To enforce constraints, the constrained distribution $q_\mathcal{C}(y_t \mid x, y_{<t})$ is given by:

$$q_\mathcal{C}(y_t \mid x, y_{<t}) \propto p_\theta(y_t \mid x, y_{<t})\, \mathbb{I}[y_t \in \mathcal{C}(y_{<t})], \tag{3}$$

where $\mathbb{I}[y_t \in \mathcal{C}(y_{<t})] = 1$ when $y_t \in \mathcal{C}(y_{<t})$, and 0 otherwise.

### 3.2 Decoding with ASGs

We instantiate $\mathcal{C}$ using ASGs, which is expressive enough to capture $\mathcal{C}_{\text{SEM}}$. Standard ASG formulation allows specification of whether a *complete* string $w \in L(G)$. To integrate ASG into decoding and enable semantic control, we have extended ASGs to enable next-token *valid completions*. We expand on this here.

#### 3.2.1 Semantically Valid Completions

When instantiating our constraint function with ASGs, valid next-token choices cannot be determined from the current parse frontier alone, since semantic constraints may encode complex, non-local relationships

and background facts. To address this, we track the complete set of partial parse trees consistent with the generated prefix and, for each candidate token, verify whether extending at least one of these trees still satisfies all ASP constraints. Thereby, admitting only tokens that preserve *some* valid partial parse, we can ensure every prefix remains on a guaranteed path to a fully derivable, hence semantically coherent output.

Formally, given a vocabulary $\Sigma$, a valid ASG is the tuple $G_{\text{ASG}} = \langle G_{CF}, \Psi \rangle$ where the terminals $T \in \Sigma$ align with LLM's vocabulary $\mathcal{V}$ (we discuss this in Section 3.2.2). For any prefix sequence $y_{<t} \in T^*$, we define $\Delta(y_{<t})$ as the set of partial parse trees rooted at a start symbol $S$ with terminal frontier $y_{<t}$. Each tree $\delta \in \Delta(y_{<t})$ must satisfy all ASP constraints $\Psi$ (i.e., $sat(G_{\text{ASG}}[\delta])$). In order to handle multiple valid parse trees for a given string, we maintain the complete set of partial trees, each representing a possible grammatical derivation of the prefix. We use $\delta \oplus a$ to denote extending a partial tree $\delta$ by token $a \in T$, and $\Delta(y_{<t} \circ a)$ to represent a collection of all partial parse trees consistent with the extended sequence $y_{<t} \circ a$. Building on these definitions, we can specify the set of valid completions:

$$\mathcal{C}_{\text{ASG}}(y_{<t}) = \{y_t \in T \mid \exists \delta \in \Delta(y_{<t}), \delta \oplus y_t \in \Delta(y_{<t} \circ y_t)\}. \tag{4}$$

The completion function $\mathcal{C}_{\text{ASG}}(y_{<t})$ essentially returns precisely those tokens $y_t$ for which at least one partial parse tree in $\Delta(y_{<t})$ can be correctly extended to yield $\Delta(y_{<t} \circ y_t)$. This procedure guarantees that every selected token $y_t \in \mathcal{C}_{\text{ASG}}(y_{<t})$ maintains $G_{\text{ASG}}[PT]$ with respect to $\Psi$ as defined in Equation (1), preserving a feasible path to a *complete* derivation in $L(G_{\text{ASG}})$.

The decoding procedure initiates with a special start symbol $S$ (corresponding to LLM's `<BOS>` token) and continues until either the only valid completion is an end symbol (`<EOS>` token), or the LLM terminates generation when `<EOS>` appears among the valid completions.

**Guaranteeing Semantic Validity** We restrict each generation step only to tokens that preserve at least one valid partial parse, the goal here is to ensure that every prefix remains extendable to a complete derivation satisfying all constraints. This invariance guarantees that once decoding terminates, the complete output is in the (ASG's) language.

More precisely, the structure of ASGs guarantees, by construction, that any complete sequence $y \sim q_{\mathcal{C}_{\text{ASG}}}$ belongs to $L(G_{\text{ASG}})$, where $q_{\mathcal{C}}$ and $\mathcal{C}_{\text{ASG}}$ are defined in Equation (3) and Equation (4) respectively. This guarantee follows from our token-level invariant: at each step $t$, selecting $y_t \in \mathcal{C}_{\text{ASG}}(y_{<t})$ ensures that at least one partial parse $\delta_{t-1} \in \Delta(y_{<t})$ can be extended to some $\delta_t \in \Delta(y_{<t+1})$ while preserving all ASP constraints. The ASP solver enforces this invariant by examining all feasible extensions at every step. When generation terminates, the resulting parse tree satisfies $sat(G_{\text{ASG}}[PT])$ by construction, guaranteeing $y \in L(G_{\text{ASG}})$. This provides stronger semantic guarantees than approaches that rely on external solvers or prompt engineering (detailed in Section 6) through aligning constraints at the token-level.

### 3.2.2 Vocabulary Alignment

Our discussion thus far has assumed perfect alignment between the ASG terminals $T$ and the LLM vocabulary $\mathcal{V}$. In practice, however, this alignment requires careful consideration as multiple LLM tokens might compose to a single terminal, and vice versa (Beurer-Kellner et al., 2024).

As such, we formalize this alignment through bidirectional mapping functions $\tau : T^* \to \mathcal{V}^*$ and $\tau^{-1} : \mathcal{V}^* \to T^* \cup \{\perp\}$. For any sequence of terminals $(a_1, \ldots, a_n) \in T^*$, $\tau(a_1, \ldots, a_n)$ produces a sequence of vocabulary tokens $(v_1, \cdots, v_m) \in \mathcal{V}^*$ that compose it, while $\tau^{-1}$ maps a sequence of tokens back to a sequence of terminals if valid, returning $\perp$[3] otherwise. These mappings must satisfy the following consistency property:

$$\forall s \in T^* : \tau^{-1}(\tau(s)) = s \quad \text{(mapping consistency)} \tag{5}$$

In practice, $\tau$ automatically *precomputes* all possible expansions of sequences in $T^*$ wrt. $\mathcal{V}$. Since in our domains $T$ is finite, enumerating valid expansions is tractable.

---

[3]representing an invalid or undefined mapping

### 3.3 Decoding with Token-Aligned Semantic Tree Search

In Section 3.1, we introduced constrained sampling, and in Section 3.2, we defined the constraint function $\mathcal{C}_{\text{ASG}}$ that guarantees semantic validity. While this guarantees that every generated sequence $y$ is semantically valid, it does not guarantee that $y$ is the correct solution to the problem. Consider the Blocksworld example in Figure 1 (a): a sequence of actions repeatedly picking up and putting down the same block would be semantically valid as it follows all the domain rules. However, it would not achieve the desired goal state.

One might consider encoding the goal-state as a requirement directly into the ASG's constraints $\Psi$, forcing only sequences terminating in the goal to be valid. However, this approach would transform `SEM-CTRL` from being *task-specific* (applicable to any instance of Blocksworld) to being *instance-specific* (tied to one particular Blocksworld configuration and goal). Instead, we want $\Psi$ to express general rules and semantic knowledge about the domain, while using a separate mechanism to achieve instance goals. We exploit a *semantic* MCTS procedure to enforce token-level control over the generation process such that it is capable of globally optimizing sequences using domain-specific rewards to search for correct solutions. This allows for explicit exploration of multiple *semantically valid* trajectories through *task-aware* scoring of the trajectories and value backpropagation to guide future expansions.

We note that our approach fundamentally differs from traditional constrained decoding approaches, which only perform local pruning of invalid tokens at each step.

#### 3.3.1 Token-level decoding as MDP

We formulate sequence generation as a Markov Decision Process (MDP), treating token selection as sequential decision-making (Zhang et al., 2023b; Wan et al., 2024). This formulation will allow us to apply principled search while maintaining semantic control through our ASGs. The MDP is defined as follows: 1) States $s \in \mathcal{S}$ represents partial generation $(x, y_{<t})$ consisting of the input prompt $x$ and the tokens generated so far $y_{<t}$; 2) Actions $a \in \mathcal{A}$ correspond to selecting tokens from the LLM vocabulary $\mathcal{V}$; and 3) Transitions $\mathcal{T} : \mathcal{S} \times \mathcal{A} \to \mathcal{S}$ that deterministically appends the selected token to the current sequence: $\mathcal{T}((x, y_{<t}), a) = (x, y_{<t} \circ a)$.

In this framework, the LLM serves as a policy $\pi_\theta : \mathcal{S} \to \mathcal{A}$ that maps each state to a distribution over actions: $\pi_\theta(a|s_t) = p_\theta(a|x, y_{<t})$. Here, the goal is thus to find a sequence $y$ that maximizes the cumulative reward. Generation continues until a maximum length is reached or EOS is generated.

**Domain Specific Reward Design** Unlike prior work that rely on LLM-based approximate rewards (Hao et al., 2023; Wan et al., 2024), we design explicit domain-specific rewards $\mathcal{R} : \mathcal{S} \times \mathcal{A} \to \mathbb{R}$. This aligns with our overarching goal of *guaranteed correctness*. Our reward function combines two key elements: 1) semantic validity (enforced by ASG constraints); and 2) task-specific distance functions. Formally, we define:

$$\mathcal{R}(s_t, a) = \begin{cases} 1 & \text{if } y_{<t} \circ a \in L(G_{\text{ASG}}) \wedge \rho(y_{<t} \circ a) = 0 \\ -\rho(y_{<t} \circ a) & \text{if } y_{<t} \circ a \notin L(G_{\text{ASG}}) \vee \rho(y_{<t} \circ a) \neq 0 \end{cases} \tag{6}$$

Here, $\rho(\cdot)$ measures the "distance to goal" and imposes penalties for invalid generations. In our empirical evaluations in Section 4, we highlight tasks where formal guarantees and exact correctness are achievable.

#### 3.3.2 Semantically guided search

Building on our MDP formulation, we exploit MCTS to compute sequences that simultaneously satisfy necessary semantic constraints $L(G_{\text{ASG}})$ and optimize the final task-specific objective. Our semantic MCTS variant modifies the standard algorithm in three key ways:

**1. Constrained Selection:** We guide node selection using constrained token distribution $q_{\mathcal{C}_{\text{ASG}}}$. For each state we select the action $a$ according to:

$$\arg\max_a Q(s_t, a) + U(s_t, a), \tag{7}$$

where $Q(s_t, a)$ are average returns and $U(s_t, a)$ is the PUCB term (Silver et al., 2017; Zhang et al., 2023b):

$$U(s_t, a) = \beta(s) \cdot q_{\mathcal{C}_{\text{ASG}}}(a \mid s_t) \cdot \frac{\sqrt{\sum_b N(s_t, b)}}{1 + N(s_t, a)} \tag{8}$$

where $N(\cdot)$ tracks visit counts and $\beta(s)$ balances exploration against exploitation.

**2. Semantic Expansion:** When expanding leaf nodes $s_t$, we exploit $\mathcal{C}_{\text{ASG}}$ to enumerate only valid next tokens. This semantic pruning yields a small set of children ($|\mathcal{C}_{\text{ASG}}(s_t)| = 1$–$15$ in our experiments), greatly reducing the branching factor compared to unconstrained methods that consider thousands of tokens.

**3. Controlled Rollouts:** We simulate rollouts by generating a complete sequence $y$ from state $s_t$ according to $y \sim \prod_t^{|y|} q_{\mathcal{C}_{\text{ASG}}}(\cdot \mid s_t)$, where $q_{\mathcal{C}_{\text{ASG}}}$ guarantees semantic validity throughout the rollout. In practice, we use beam search with beam size=1 or greedy decoding for efficient and deterministic completions.

This semantic guidance distinguishes `SEM-CTRL` from previous search-guided (reasoning-related and other) approaches (detailed in Section 6). Our strategy of constraining exploration to valid trajectories and using domain-specific rewards helps ensure LLM's search process remains within the space of meaningful solutions, providing reliable control when coupled with ASGs.

### 3.4 Handling Computational Overheads

While our semantically constrained MCTS approach provides robust control over generation, it introduces additional computational costs. We address this through three complementary optimization strategies:

**(1) Caching Partial ASG Derivations.** We cache partial parse trees $\delta$ to avoid recomputing $\mathcal{C}_{\text{ASG}}(y_{<t})$.

**(2) Semantic Tree Pruning** Unlike methods that rely on arbitrary top-$k$ sampling (Zhang et al., 2023b; Wan et al., 2024; Hao et al., 2023), our semantic constraints enable effective deep search by maintaining a small branching factor. As detailed in Section 3.3.2, $\mathcal{C}_{\text{ASG}}$ enables exploration at significant depths (e.g., 256 tokens) while preserving both domain feasibility and solution reachability—if a correct solution exists at depth $d$, it remains discoverable. This allows reliable traversal to high-reward solutions.

**(3) Tree-Structure Caching** Following Zhang et al. (2023b), we exploit the hierarchical nature of MCTS by caching tree structures and sequences. For any prefix $y_{<t}$ that reappears across iterations, we reuse its stored top-$k$ expansions and rollouts. Similarly, we cache partial or complete rollouts to avoid redundant sub-tree computations and duplicate model calls. This ensures we only pay for expansions *once per node*.

## 4 Experiments

We evaluate `SEM-CTRL` across four diverse classes of tasks to assess its effectiveness in enforcing complex constraints while generating correct solutions. Our evaluation spans Synthetic Grammar Synthesis (SGS), Combinatorial Reasoning (CR), JSON parsing, and Planning tasks.[4]

### 4.1 Task Setup

We describe the details of each class of tasks below.

**Synthetic Grammar Synthesis** We evaluate `SEM-CTRL` on three canonical SGS tasks: $L_1 = \{a^n b^n c^n \mid n \geq 1\}$, $L_2 = \{a^m b^n c^m d^n \mid m, n \geq 1, m \neq n\}$, and $L_{\text{copy}} = \{ww \mid w \in \{a, b\}^+\}$. For each task, we evaluate over 30 samples. For $L_1$, we linearly increment $n$ from 1 to 30.[5] We define the distance function $\rho_1 = n - n_w$, where $n_w$ represents the count of $n$ in the generated string $y$. For $L_2$, we increment $m + n$ from 3 to 32 and use distance function $\rho_2 = (m+n) - (m_w + n_w)$, where $m_w$ and $n_w$ are the respective counts in $y$. For $L_{\text{copy}}$, we divide the 30 samples into 10 equal parts across three evaluation criteria: incrementing $a$ from 1 to 10, incrementing $b$ from 1 to 10, and maximizing their product. Here, $\rho_3$ measures the distance to these targets.

---

[4]See Appendix C for detailed task descriptions, constraints, and distance functions $\rho(\cdot)$.
[5]In fact, our choice of 30 is due in part to Allen-Zhu & Li (2024)'s work.

Each task involves prompting an LLM with exemplars $E = e_1, \cdots, e_l$ where $l \in [0, 5]$, with the number of exemplars varying based on the complexity of the target string (e.g., fewer examples for simpler cases).

**Combinatorial Reasoning** The combinatorial reasoning related tasks include benchmarks from prior work (Long, 2023; Borazjanizadeh et al., 2024; Seely et al., 2025), these include: (1) Sudoku 3x3 boards, (2) Sudoku 4x4 boards, and (3) 3-Graph Coloring (NP-complete) (Law et al., 2019). For Sudoku, we use the dataset from Long (2023) with one-shot prompts, where we expect the solutions in a nested list format (as shown in Table 7 in the Appendix). The MCTS employs a sparse reward function where $\rho(\cdot) = 0$ for solved boards and 1 otherwise. We further note that our ASG only encodes game rules.

For 3-Graph Coloring, we prompt the LLM to generate valid colorings for graphs with at most 5 nodes and between 3 and 10 edges, using $\rho(\cdot) = e - e_w$ as the distance function. Following the SGS setup, we use few-shot prompting and require solutions to be formatted as edge lists $(i, j)$.

**Planning** Our benchmark comprises the Blocksworld domain using PlanBench's plan generation task (Valmeekam et al., 2024). The dataset contains 600 one-shot problems requiring the LLM to provide action sequences from an initial state to the goal. In our cases, we expect the LLM to return a structured output: comma-separated action sequences. Our ASG constraints ($\Psi_{PR}$) ensure action preconditions are satisfied, while $\Psi_B$ tracks state information and general rules (e.g., goal completion termination). The distance function $\rho(\cdot)$ combines a heuristic function $h(s, g)$ approximating goal distance (i.e., relaxed plan heuristic given by Pyperplan Alkhazraji et al. (2020)) with a plan length penalty $\alpha \cdot \text{len(plan)}$ discouraging repetitive actions.

**Parsing** Our final benchmark is a parsing task using the dataset from NousResearch (2024), where the LLM extracts information from natural language and formats it as JSON according to a schema. While `SEM-CTRL` is designed for tasks with well-defined semantics and a reward function capturing correctness, some structured generation and semantic parsing tasks lack these properties (Lei et al., 2025; Roy et al., 2023; Wang et al., 2023, inter alia). We include this task as a representative example to demonstrate `SEM-CTRL`'s broader applicability. Here, we run `SEM-CTRL` without MCTS and use an ASG encoding a CFG, demonstrating semi-open-ended structural generation capability without semantic constraints.

## 4.2 Experimental Setup

**Models and Baselines** We evaluate `SEM-CTRL` using three variants of Llama 3: Llama 3.2 1B, Llama 3.1 8B, and Llama 3.1 70B, and three state-of-the-art reasoning models: o4-mini, DeepSeek-R1, and o1-preview. To disentangle the contributions of different components, we systematically evaluate combinations of sampling algorithms with varying constraint types. We describe the key baselines below:

- **Base Unconstrained:** Greedy sampling to assess the base model's capability.

- **Base with $\mathcal{C}_{\text{CFG}}$:** Locally constrained syntactic decoding. This approach is analogous to various work in controlled decoding where the LLM's next tokens are masked according to CFG constraints (Geng et al., 2023; Ugare et al., 2024; Beurer-Kellner et al., 2024, inter alia), though we implement this through an ASG encoding a CFG.

- **XGrammar (Dong et al., 2024):** While Base with $\mathcal{C}_{\text{CFG}}$ is functionally equivalent to prior work in syntactic control, we run an additional baseline for the parsing task for completeness and to empirically highlight this equivalence.

- **Base with $\mathcal{C}_{\text{CSG}}$:** We mask LLM logits with terminals from an ASG encoding context-sensitive and semantic constraints. This parallels work in semantic parsing, though prior methods typically use CFG with ad-hoc constraints (e.g., Scholak et al., 2021; Poesia et al., 2022; Roy et al., 2023).

- **BoN Unconstrained:** Best-of-N (BoN) serves as a simple constraint satisfaction mechanism by sampling $N$ generations and rejecting invalid samples according to $\mathcal{C}$ and ranking solutions by $\mathcal{R}(\cdot)$ (Welleck et al., 2024). We set $N$ to match `SEM-CTRL`'s computational budget (maximum number of MCTS samples generated during search) for fair comparison. See Appendix F for $N$ values.

- **MCTS Unconstrained:** MCTS applied at the token level, corresponding to a range of search-guided reasoning approaches (e.g., Zhang et al., 2023b; Wan et al., 2024).

We note that several of our baselines are themselves either exact or approximate global correctness optimization methods in the sense that they seek high-reward outputs at inference time (e.g., unconstrained MCTS via principled search (Zhang et al., 2023b; Hao et al., 2023; Wan et al., 2024)). Throughout the main text, we compare against two primary baselines: (1) Base unconstrained; (2) Unconstrained BoN sampling. In Section 5.3, we present a comprehensive ablation study on the Blocksworld benchmark with all algorithm-constraint combinations, and provide the full results for the other tasks in Appendix H. The complete list of all baseline methods is detailed in Appendix E.[6]

For BoN, we use temperature-1 nucleus sampling with standard parameters (top-$p = 1.0$, top-$k = 50$). For `SEM-CTRL`, we set top-$k$ to the number of terminals in the grammar. For the reasoning models, we use Microsoft Azure's API. We average all results over 3 runs[7].

**Evaluation Metrics** We assess performance using three primary evaluation metrics: (1) **A**: Measures task-specific accuracy (correctness); (2) $\mathbf{V_{CFG}}$: Measures CFG validity ($y \in L(G_{\text{CFG}})$); (3) $\mathbf{V_{CSG}}$: Measures CSG validity ($y \in L(G_{\text{CSG}})$); (4) $\mathbf{V_{SEM}}$: Measures semantic validity ($y \in L(G_{\text{SEM}})$).

## 5 Results

We present an empirical evaluation of `SEM-CTRL` on the experimental set-up introduced in Section 4. Our analysis is structured to answer several key research questions, which we address in the following subsections.

### 5.1 Overall Results

We present our main findings in Table 1. We observe that `SEM-CTRL` consistently matches or exceeds baseline performance across all tasks. We highlight the following key observations:

Table 1: Accuracy (A) results for the tasks: $a^n b^n c^n$, $a^m b^n c^m d^n$, Copy, Graph Coloring (Graph), Sudoku 3x3 (Sudoku-3), Sudoku 4x4 (Sudoku-4), and Planning (Blocks), using various sampling algorithms (Alg.), with different base LLMs (Model).

| Alg. | Model | Synthetic Grammar Synthesis | | | Combinatorial Reasoning | | | Planning |
|---|---|---|---|---|---|---|---|---|
| | | $a^n b^n c^n$ | $a^m b^n c^m d^n$ | Copy | Graph | Sudoku-3 | Sudoku-4 | Blocks |
| Base | Llama 1B | 3.3% | 0.0% | 10.0% | 0.0% | 0.0% | 0.0% | 0.0% |
| | Llama 70B | 30.0% | 0.0% | 60.0% | 37.5% | 90.0% | 30.0% | 23.2% |
| BoN | Llama 1B | 7.8% | 1.1% | 48.9% | 0.0% | 0.0% | 0.0% | 4.3% |
| | Llama 70B | 71.1% | 22.2% | 88.9% | **100.0%** | 96.7% | 80.0% | 48.8% |
| API | o1-preview | 83.3% | 80.0% | 96.7% | 75.0% | **100.0%** | **100.0%** | 94.5% |
| | DeepSeek-R1 | 83.3% | 70.0% | 96.7% | 75.0% | **100.0%** | **100.0%** | **96.5%** |
| | o4-mini | 93.3% | 93.3% | **100.0%** | 75.0% | **100.0%** | **100.0%** | **98.5%** |
| `SEM-CTRL` | Llama 1B | **100.0%** | **100.0%** | **100.0%** | **100.0%** | **100.0%** | **100.0%** | 74.0% |
| | Llama 70B | **100.0%** | **100.0%** | **100.0%** | **100.0%** | **100.0%** | **100.0%** | **96.8%** |

**Parameter Efficiency** `SEM-CTRL` with Llama 1B consistently outperforms both greedy and BoN Llama 70B variants across all tasks. Prominently, in the complex $a^m b^n c^m d^n$ task, `SEM-CTRL` with Llama 1B achieves 100% accuracy while Llama 70B fails completely (0% accuracy). This indicates that our semantic control framework effectively compensates for model size, enabling smaller models to solve complex reasoning tasks through guided exploration.

**Comparison to State-of-the-Art** `SEM-CTRL` achieves superior or competitive performance compared to current state-of-the-art reasoning models, including o4-mini, DeepSeek-R1, and o1-preview, across SGS, Combinatorial Reasoning, and Planning domains. On SGS and Combinatorial Reasoning tasks, `SEM-CTRL`

---

[6]Code will be released upon acceptance of the paper or once the review process has concluded, in order to preserve anonymity.
[7]except in Blocksworld due to computational cost and our inference budget constraints.

consistently outperforms all reasoning models. For instance, in $a^n b^n c^n$, `SEM-CTRL` achieves 100% accuracy compared to o1-preview's 83.3% and o4-mini's 93.3%. The performance gap becomes more pronounced on the more complex $a^m b^n c^m d^n$ task, where performance degrades for o1-preview (80.0%) and DeepSeek-R1 (70.0%), while o4-mini maintains 93.3% and `SEM-CTRL` reaches perfect accuracy. The advantage is particularly notable in Graph Coloring, a complex NP-complete problem, where all reasoning models achieve only 75% accuracy while `SEM-CTRL` maintains 100% accuracy. This demonstrates that our semantic constraints and guided search provide reliable, systematic, and consistent reasoning capabilities for structured problems, ensuring semantic validity and strong empirical performance across these diverse domains.

**Task Complexity** The effectiveness of `SEM-CTRL` is particularly evident in Blocksworld planning, arguably our most complex task requiring long-horizon reasoning, precise state tracking, and satisfying action pre- and post-conditions across 600 samples. Even with the smaller Llama 1B model, `SEM-CTRL` achieves 74% accuracy, outperforming larger closed-source models including Claude 3.5 Sonnet (57.6%) and GPT-4o (28.3%) (Valmeekam et al., 2025). Statistical analysis using paired permutation tests ($\alpha = 0.05$) reveals that performance differences between `SEM-CTRL` with Llama 70B (96.8%) and o4-mini (98.5%) or DeepSeek-R1 (96.5%) are not statistically significant, though the improvement over o1-preview (94.5%) is.

Notably, `SEM-CTRL` achieves this superior or comparable performance to state-of-the-art reasoning models despite the complexity of this task purely at inference time with off-the-shelf LLMs. This competitive performance comes with the added benefit of guaranteed semantic validity (100% constraint satisfaction), whereas reasoning models cannot ensure constraint adherence. This demonstrates `SEM-CTRL`'s effectiveness in achieving reliable and consistent inference with semantic guarantees across complex reasoning tasks.

**Specializing LLM models** One general pattern that we observe in our experiments is that `SEM-CTRL` is capable of transforming general-purpose off-the-shelf LLMs into domain-specialized models at inference time with guarantees on obtaining the correct and semantically valid solutions.

## 5.2 Can `SEM-CTRL` guarentee grammatical validity?

To evaluate how well `SEM-CTRL` and unconstrained LLMs capture control levels, we follow prior work (Schucher et al., 2022; Drozdov et al., 2023; Levy et al., 2023, especially) and present $V_{CFG}$ and $V_{CSG}$ scores on SGS in Table 2. The results reveal the unreliability of LLMs, where unconstrained models fail to adhere to constraints consistently; `SEM-CTRL` addresses this by guaranteeing validity at inference time.

Two key patterns emerge: First, all models struggle notably more with $V_{CSG}$ than $V_{CFG}$—small models like Llama 1B achieve 79% $V_{CFG}$ but only 23% $V_{CSG}$, while large models like Llama 70B achieve 99% $V_{CFG}$ vs. 39% $V_{CSG}$. This contrasts with prior work's focus on syntax alone and motivates more expressive controls. Second, while sampling algorithms like BoN improve performance (Llama 70B with BoN reaches 79% $V_{CSG}$ and 35% for 1B), none guarantee perfect validity.

Table 2: Accuracy (A), context-free ($V_{CFG}$) and context-sensitive ($V_{CSG}$) correctness results on SGS

| Alg. | Model | A | $V_{CFG}$ | $V_{CSG}$ |
|---|---|---|---|---|
| Base | Llama 1B | 4.4% | 79.0% | 23.0% |
| | Llama 70B | 30.0% | 99.0% | 39.0% |
| BoN | Llama 1B | 19.3% | 65.0% | 35.0% |
| | Llama 70B | 60.7% | 97.0% | 79.0% |
| API | o1-preview | 86.7% | **100.0%** | 88.0% |
| | DeepSeek-R1 | 83.3% | **100.0%** | 87.0% |
| | o4-mini | 94.7% | 99.0% | 95.0% |
| SEM-CTRL | Llama 1B | **100.0%** | **100.0%** | **100.0%** |
| | Llama 70B | **100.0%** | **100.0%** | **100.0%** |

Even state-of-the-art reasoning models fall short: o1-preview and DeepSeek-R1 achieve only 88% and 87% $V_{CSG}$ respectively, while o4-mini reaches 95%. This reveals two failure modes for accuracy: inability to capture constraints and failure to ensure solution correctness. These empirical observations suggest that while LLMs can approximate constraints, they lack robustness. `SEM-CTRL` instead achieves perfect $V_{CFG}$ and $V_{CSG}$ across tasks and model sizes, demonstrating explicit semantic control is crucial for reliable generation.

### 5.3 How does each disentangled dimension contribute to SEM-CTRL's abilities?

We now present the results of an ablation study for SEM-CTRL on Blocksworld planning in Table 3 to disentangle the contributions of each dimension (semantic control, semantic search, and parameter scale). Following Valmeekam et al. (2023), we subsample 50 problems and systematically evaluate constraint types (unconstrained, $\mathcal{C}_{\text{CFG}}$, $\mathcal{C}_{\text{SEM}}$) across sampling algorithms (Base, BoN, and MCTS).

Key empirical findings emerge: First, incrementally moving from no control to syntactic to semantic constraints shows consistent improvements—most pronounced with BoN where Llama 1B improves from 6% to 56%. Second, $\mathcal{C}_{\text{CFG}}$ only benefits when combined with MCTS, yet still underperforms BoN with $\mathcal{C}_{\text{SEM}}$. Across all configurations, we find that $\mathcal{C}_{\text{SEM}}$ provides the most substantial gains.

Our SEM-CTRL strategy, (which essentially is $\mathcal{C}_{\text{SEM}}$ + MCTS) achieves substantially better results than either component alone, enabling even the Llama 1B model to outperform baselines and the 70B model to match or exceed reasoning model performance. This highlights the importance of semantic control, which fundamentally provides crucial structure for MCTS to effectively navigate the solution space, producing improvements exceeding the sum of their individual contributions. We present ablations on SGS and CR in Appendix H, where we observe similar trends.

Table 3: Results from Blocksworld ablation study

| Alg. | Model | $\mathcal{C}$ | A | $\mathbf{V_{CFG}}$ | $\mathbf{V_{SEM}}$ |
|---|---|---|---|---|---|
| Base | Llama 1B | - | 0% | 100% | 0% |
| | Llama 1B | $\mathcal{C}_{\text{CFG}}$ | 0% | 100% | 0% |
| | Llama 1B | $\mathcal{C}_{\text{SEM}}$ | 6% | 100% | 100% |
| | Llama 70B | - | 30% | 100% | 40% |
| | Llama 70B | $\mathcal{C}_{\text{CFG}}$ | 30% | 100% | 40% |
| | Llama 70B | $\mathcal{C}_{\text{SEM}}$ | 40% | 100% | 100% |
| BoN | Llama 1B | - | 6% | 94% | 6% |
| | Llama 1B | $\mathcal{C}_{\text{CFG}}$ | 6% | 100% | 6% |
| | Llama 1B | $\mathcal{C}_{\text{SEM}}$ | 56% | 100% | 100% |
| | Llama 70B | - | 66% | 100% | 66% |
| | Llama 70B | $\mathcal{C}_{\text{CFG}}$ | 66% | 100% | 66% |
| | Llama 70B | $\mathcal{C}_{\text{SEM}}$ | 90% | 100% | 100% |
| MC-TS | Llama 1B | - | 12% | 100% | 40% |
| | Llama 1B | $\mathcal{C}_{\text{CFG}}$ | 38% | 100% | 66% |
| | Llama 70B | - | 46% | 96% | 68% |
| | Llama 70B | $\mathcal{C}_{\text{CFG}}$ | 62% | 100% | 92% |
| SEM-CTRL | Llama 1B | $\mathcal{C}_{\text{SEM}}$ | 76% | 100% | 100% |
| | Llama 70B | $\mathcal{C}_{\text{SEM}}$ | **98%** | 100% | 100% |

### 5.4 How computationally efficient is SEM-CTRL compared to reasoning models?

We begin by formally decomposing SEM-CTRL's wall-clock time and computational cost into its constituent components as follows:

$$T_{\text{total}} \approx (N_{\text{tokens}} \times T_{\text{LLM}}) + T_{\mathcal{C}} + T_{\text{search\_ops}} \tag{9}$$

where $N_{\text{tokens}}$ is the total number of generated tokens, $T_{\text{LLM}}$ is the hardware-dependent latency of a single forward pass, $T_{\mathcal{C}}$ is the time spent evaluating ASG constraints, and $T_{\text{search\_ops}}$ represents MCTS operations (e.g., incrementing visit counts). Notably, $T_{\text{search\_ops}}$ consists of constant-time CPU operations, making its contribution negligible compared to $T_{\mathcal{C}}$ or the dominant GPU costs of $T_{\text{LLM}}$.

We analyze computational and token costs compared to reasoning models to evaluate SEM-CTRL's efficiency established via well-structured search, focusing on hardware-independent metrics that enable fair comparisons. The class of reasoning models is understood to perform a form of reasoning via *reasoning tokens* before outputting the final generation. Following recent work in token-level search (e.g., Valmeekam et al., 2025; Zhou et al., 2024), we report $N_{\text{tokens}}$ and $T_{\mathcal{C}}$ as our primary metrics. Since our baselines are closed-source reasoning models running on unknown hardware and inference strategies, token counts provide a hardware-independent proxy for inference cost. This naturally enables comparison with reasoning models: just as reasoning models consume tokens during their reasoning process, SEM-CTRL consumes tokens during search.

Table 4: Average computational efficiency per sample on SGS, CR, and Blocksworld Planning (Blocks) using Llama 70B

| Class | Alg. | $N_{\text{tokens}}$ | $T_{\mathcal{C}}$ |
|---|---|---|---|
| SGS. | o1-preview | 1639.0 | - |
| | DeepSeek-R1 | 927.3 | - |
| | o4-mini | 705.3 | - |
| | SEM-CTRL | **250.2** | 123.24 |
| CR. | o1-preview | 3184.8 | - |
| | DeepSeek-R1 | 3016.5 | - |
| | o4-mini | 1190.5 | - |
| | SEM-CTRL | **123.3** | 15.90 |
| Blocks | o1-preview | 2457.9 | - |
| | DeepSeek-R1 | 2345.8 | - |
| | o4-mini | 1544.5 | - |
| | SEM-CTRL | **589.3** | 35.97 |

While `SEM-CTRL` introduces constraint checking overhead ($T_C$), the results demonstrate a significant reduction in total token generation. We observe that across all tasks, `SEM-CTRL` reduces token usage by an order of magnitude. For example, we see in CR tasks that `SEM-CTRL` is 25.8×, 24.5×, and 9.7× more efficient than o1-preview, DeepSeek-R1, and o4-mini, respectively, while achieving perfect accuracy. $T_C$ varies by task structure, with SGS showing higher overhead (123s) due to its synthetic nature requiring notably deeper parse trees (up to depth 64) compared to shallower structures in CR and Planning. We note that this computational time depends on the depth of the sequences' parse trees in the grammar, which, for this task, is stressed to be higher than prior work (e.g., Allen-Zhu & Li, 2024).

### 5.5 How well does `SEM-CTRL` work without semantics and rewards?

We turn to assessing `SEM-CTRL`'s broader applicability when stripped of semantic structures and reward signals via the JSON parsing results in Table 5. Here, `SEM-CTRL` operates under Base sampling and achieves perfect CFG validity (100%) across model sizes, matching XGrammar (Dong et al., 2024), which we include as a representative baseline of existing CFG-constrained methods (see Sections 4.2 and 6). As anticipated, both approaches achieve identical results since they are functionally equivalent when only CFG constraints are present. Large models like Llama 70B (96.9%) and state-of-the-art reasoning models (98.4%) fail to guarantee perfect syntactic validity on this widely-used structured generation task. This confirms `SEM-CTRL`'s reliable constraint enforcement across the spectrum, from complex semantic reasoning tasks to basic syntactic generation.

Table 5: $V_{CFG}$ results on the JSON parsing task

| Alg. | Model | $V_{CFG}$ |
|---|---|---|
| Base | Llama 1B | 75.0% |
| | Llama 70B | 96.9% |
| Xgrammar | Llama 1B | **100.0%** |
| | Llama 70B | **100.0%** |
| API | DeepSeek-R1 | 98.4% |
| | o4-mini | 98.4% |
| `SEM-CTRL` | Llama 1B | **100.0%** |
| | Llama 70B | **100.0%** |

### 5.6 Does fine-tuning impact models' performance with `SEM-CTRL`?

While `SEM-CTRL` is designed as an inference-time algorithm enabling strong performance without the high training costs of fine-tuning, comparing the two approaches offers a valuable perspective on the trade-offs and potential complementarity between inference-time search and training-time adaptation. We conduct experiments on Blocksworld, fine-tuning Llama-1B models using varying fractions of the PlanBench training data (5%, 10%, 20%, 50%, and 100%). For each fine-tuned model, we compare greedy decoding against `SEM-CTRL`. We evaluate on the Blocksworld ablation set introduced in Section 5.3 and report (1) **Accuracy (A)**: percentage of problems correctly solved, and (2) **Sample Efficiency (Seq.)**: the average number of full sampled sequences explored by MCTS before finding a correct solution, with results summarized in Table 6.

Table 6 reveals three key findings. First, without fine-tuning, `SEM-CTRL` with Llama-1B substantially outperforms fine-tuned greedy baselines across all data regimes (76% vs. 16% maximum greedy accuracy), confirming that `SEM-CTRL` alone provides strong inference-time gains that fine-tuning does not reliably recover. Second, fine-tuning and `SEM-CTRL` are complementary: as the amount of fine-tuning data increases, accuracy improves further (up to 90%), while the number of sampled sequences required by MCTS drops

Table 6: Accuracy (A) and sample efficiency (Seq.) results on the Blocksworld ablation set with varying Fine-Tuning (FT) data splits on `SEM-CTRL` and greedy sampling.

| Model | FT Data | A | Seq. |
|---|---|---|---|
| Llama 1B (Greedy) | 0% | 0.0% | – |
| | 5% | 12.0% | – |
| | 10% | 10.0% | – |
| | 20% | 16.0% | – |
| | 50% | 16.0% | – |
| | 100% | 14.0% | – |
| Llama 1B (SEM-CTRL) | 0% | 76.0% | 71.14 |
| | 5% | 58.0% | 53.80 |
| | 10% | 76.0% | 46.00 |
| | 20% | 90.0% | 30.04 |
| | 50% | 88.0% | 17.44 |
| | 100% | 88.0% | **13.22** |
| Llama 70B (SEM-CTRL) | 0% | **98.0%** | 19.12 |

by up to 5.4× (71.14 vs. 13.22 sequences with full data). This indicates that fine-tuning helps shape the model's distribution by learning to generate more valid trajectories that `SEM-CTRL` can efficiently explore via robust semantic control to ensure correctness.

Finally, regarding parameter efficiency, Llama 1B with `SEM-CTRL` and moderate fine-tuning (20–50% of FT data) rivals the performance of Llama 70B, despite being orders of magnitude smaller and 1.4× more sample efficient (13.22 vs. 19.12 sequences). This demonstrates that `SEM-CTRL` significantly reduces the amount of

fine-tuning required to reach a target performance level. Ultimately, our results clarify that `SEM-CTRL` serves as a versatile inference-time mechanism rather than a strict alternative to fine-tuning: it ensures robust performance when training costs are prohibitive, yet yields synergistic gains when fine-tuning is feasible, simultaneously improving both accuracy and search efficiency.

## 6 Related Work

**Controlled Decoding** Research in controlled decoding has evolved to increasingly expressive forms of control. Early work focused on simple lexical constraints (Welleck et al., 2024; Hokamp & Liu, 2017; Anderson et al., 2017, inter alia), subsequently formalized using predicate logic (Lu et al., 2022; Zhang et al., 2023a) and extended to higher abstraction levels (Lew et al., 2023; Yao et al., 2024a). More sophisticated approaches exploit CFGs to enforce syntactic validity (Beurer-Kellner et al., 2023; Koo et al., 2024; Ugare et al., 2024, inter alia). Prior work refers to constrained decoding as control via hard constraints and controlled decoding via preferences (Yang & Klein, 2021; Meng et al., 2024). We refer to both as control, formalized in Section 3.1.

**Semantic Parsing with LLMs** LLM-based semantic parsing has progressed from unconstrained methods via fine-tuning (inter alia Roy et al., 2023; Li et al., 2021) and prompting (Schucher et al., 2022; Drozdov et al., 2023) to constrained generation with domain-specific rules. Systems like PICARD (Scholak et al., 2021) combine CFGs with domain-specific guards for SQL generation, with others extending this to other domains (Ugare et al., 2025; Shin et al., 2021; Poesia et al., 2022). In contrast, Loula et al. (2025) incorporates real-world semantic signals instead of constraints alongside CFGs or assumes LLMs can approximate CFGs. We address these limitations through ASGs, which expresses a hierarchy of constraints, guaranteeing validity across domains. `SEM-CTRL` distinguishes itself by directly employing the CSG formalism (to our knowledge, first to do so) and focusing on tasks where correctness is observable at inference, unlike other semantic parsing work where semantic constraints, correctness observability, or both are absent (i.e., JSON parsing).

**LLMs-Based Reasoning, Search, and Planning** Work in augmenting reasoning capabilities of LLMs have extended Chain-of-Thought (Wei et al., 2022) to non-linear reasoning through sequence-level tree search like Tree-of-Thought (Yao et al., 2024b) and MCTS (e.g., Murthy et al., 2024), enabling better exploration of solution paths. Token-level search methods (Zhang et al., 2023b; Wan et al., 2024) offer more granular control. However, both limit tree width, risking the exclusion of valid solutions. Others augment LLMs with tools (Qin et al., 2024) or solvers (e.g., Yang et al., 2025; Xu et al., 2024), introducing potential translation errors (Feng et al., 2024). In planning domains, reasoning models show impressive results (Valmeekam et al., 2025), yet cannot guarantee plan validity, leading to methods using search-guided reasoning (Hao et al., 2023), verifiers (Kambhampati et al., 2024), or LLMs as heuristics (Wang et al., 2023). `SEM-CTRL` addresses this via token-aligned semantic constraints with MCTS, efficiently exploring the only valid solution spaces.

## 7 Conclusion

We presented `SEM-CTRL`, a unified framework enabling controlled generation from LLMs, combining semantic constraints expressed via ASGs with token-level MCTS. We show that combining expressive semantic constraints with guided search guarantees semantic correctness and augments the capabilities of off-the-shelf LLMs. Especially, we show that it enables small models to outperform state-of-the-art LLMs (i.e., o4-mini). Our empirical results demonstrate that rich and expressive control mechanisms can transform general-purpose LLMs into robust and reliable domain-specialized models while greatly reducing token overhead.

## Broader Impact Statement

`SEM-CTRL` enables smaller, more efficient models to achieve reliable performance with formal correctness guarantees, reducing computational costs and environmental impact while democratizing access to high-quality structured generation without fine-tuning.

However, several limitations warrant consideration. `SEM-CTRL` requires that the underlying language model's learned representations align with the target constraint vocabulary, i.e., domains where the model lacks sufficient exposure to relevant terminals may exhibit degraded performance. Our approach currently operates

exclusively on text, though we hypothesize extensibility to multimodal scenarios; validation requires additional research, experimentation, and benchmarks for multimodal constraint enforcement. Furthermore, our constraint framework operates under a binary validity paradigm, precluding applications required for NLP tasks such as text summarization.

Finally, systematic constraint enforcement may amplify distributional biases present in constraint formulations, necessitating careful specification design to mitigate potential discriminatory outcomes. However, while `SEM-CTRL` does not inherently prevent such bias, and recognizing that formal correctness does not imply ethical or social appropriateness, our framework mitigates these risks through transparency. Since `SEM-CTRL` relies on symbolic constraint specifications, these rules are inherently interpretable and provide a natural mechanism to verify or audit for potential malware or malicious intent. This stands in contrast to knowledge distilled in opaque model parameters. Although such checks require ASP expertise, LLMs can be utilized to generate these constraints, thereby facilitating the democratization of auditable, controlled generation (e.g., Alviano et al., 2025; Schrader et al., 2025; Ishay et al., 2023). These limitations, while crucial, remain out of the scope of this paper, necessitating additional research in the field of controlled decoding.

### Acknowledgments

We thank Anthony Cohn and Microsoft Research's Accelerating Foundation Models Research program for the provision of Azure resources to run some of the LLMs used in the experiments in this paper. This work was supported in part by the Alan Turing Institute under Fundamental Research (Project No. PP00029) and projects 3928 and EP/Y037421/1 funded by the UKRI.

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

# A  Prompt Examples

---

**System Message:**

You are an expert in reasoning, solving puzzles, and formal languages, specifically, Context-Free and Context-Sensitive Grammars. Given a puzzle or a reasoning problem, you can easily solve it, even those requiring combinatorial reasoning. Furthermore, you can read and understand grammars, and given a grammar specification, you can generate words that consistently conform to the grammar, its language, and rules without a single mistake. Specifically, you are an expert in solving Sudoku puzzles. The generalised rules of an $n \times n$ Sudoku board are as follows:

1. Fill an $n \times n$ grid with numbers 1 through $n$.
2. Each row must contain numbers 1-$n$ without repeating.
3. Each column must contain numbers 1-$n$ without repeating.
4. Each $\sqrt{n} \times \sqrt{n}$ box must contain numbers 1-$n$ without repeating (only applicable in the case where $n$ is a perfect square).
5. Pre-filled numbers cannot be changed.

For each message, you will be presented with a Sudoku board, and you must return a solution to the puzzle conforming to its grammar. The grammar for Sudoku is as follows: `[row_1,...,row_n]`, where each `row_i` is a list of numbers separated by a comma without any spaces representing a row in the Sudoku board, i.e., `[j,...,n]`. Furthermore, separate each row by a comma without any spaces. Simply put, this grammar is a matrix (nested list format) representation of the Sudoku board. Missing numbers in the partial boards presented are represented by `*`. You must leave the pre-filled numbers unchanged. Your goal is to solve the Sudoku board and only return a valid solution to the puzzle according to the grammar; do not generate any additional text beyond the solution.

**Example Interaction:**

*User:* Generate a valid solution to the Sudoku board `[[*,3,1],[*,2,3],[3,*,2]]` where * represents a cell to be filled in. Please return your solution according to the grammar for Sudoku.
*Assistant:* `[[2,3,1],[1,2,3],[3,1,2]]`
*User:* Generate a valid solution to the Sudoku board `[[1,*,*],[*,1,*],[*,2,*]]` where * represents a cell to be filled in. Please return your solution according to the grammar for Sudoku.
*Assistant:* `[[1,3,2],[2,1,3],[3,2,1]]`

---

Figure 3: Prompt template for the 3×3 Sudoku task, showing the system message that defines the task and grammar, followed by example interactions.

Figure 3 showcases an example prompt from our tasks, specifically, the Sudoku-3×3 prompt. It adopts a standard prompting strategy. We find that a combination of a textual description of the task, syntax, and constraints, and a simplified grammar format provides better results as opposed to simply providing the entire grammar itself (i.e., in ASG or Extended Backus–Naur Form). This is in contrast with prior work (Wang et al., 2023), which found providing the complete grammar to the LLM to be helpful in the case of GPT-4. This was not the case with our experiments. We empirically choose these prompts according to the baselines' accuracies (i.e., Greedy or BoN), as we find `SEM-CTRL` to be robust to slight variations in the prompt. SGS and Combinatorial Reasoning follow a similar prompting strategy. Hence, we omit them.

We further provide the prompts used for our plan generation PlanBench task in Figure 4, given we slightly modify the task from the original dataset Valmeekam et al. (2024). Our slight changes were made by empirically examining the model's performance when required to return the action sequences as a list of actions as opposed to free-form text.

**System Message:**
You are an expert planner, where given a domain and an instance of a problem, you must provide a sequence of actions taking the agent in the environment from the initial state to the goal state in the most optimal manner. You do not select invalid actions, you do not generate sub-optimal plans, and you must conform to the domain's specifications as provided to you. In your response, you must only provide a sequence of actions separated by commas achieving the goal state without providing any other information. You must only use the actions and environment objects available to you and nothing else. Specifically, you are an expert PDDL planner in the Blocksworld domain, where given a set of blocks arranged in an initial configuration, you must provide a sequence of actions achieving the goal of arranging the blocks in a desired configuration. You are very good at this task.

**Initial and goal states are defined by a series of predicates, described below:**

1. `on BLOCK_NAME_1 BLOCK_NAME_2`: The block BLOCK_NAME_1 is on top of block BLOCK_NAME_2
2. `ontable BLOCK_NAME`: The block BLOCK_NAME is on the table
3. `clear BLOCK_NAME`: The block BLOCK_NAME is clear, i.e., no other blocks are on top of it and it is not being held
4. `handempty`: The robotic hand is not holding any blocks and is free

**The restrictions of the actions are as follows:**

1. You can only pickup or unstack one block at a time
2. You can only pickup or unstack a block if the robotic hand is empty
3. You can only pickup a block if the block is on the table and the block is clear
4. You can only unstack a block from on top of another block if the block you are unstacking was really on top of the other block
5. You can only unstack a block from on top of another block if the block you are unstacking is clear
6. Once you pickup or unstack a block, the robotic hand is holding the block
7. You can only putdown a block that the robotic hand is holding
8. You can only stack a block on top of another block if the robotic hand is holding the block being stacked
9. You can only stack a block on top of another block if the latter block is clear
10. Once you putdown or stack a block, the robotic hand becomes empty
11. Once you stack a block on top of a second block, the second block is no longer clear
12. You can only terminate the plan when the goal state is reached or your plan is complete

Block names are defined by colors, as will be shown in the specific instances of the problem. To provide a sequence of actions, you must separate them by a comma.

**Example Interaction:**
*User:* Given an instance of the Blocksworld domain as follows:
Block Objects: red, blue, orange, yellow
Initial State: clear red, clear blue, clear yellow, handempty, on blue orange, ontable red, ontable orange, ontable yellow
Goal State: on orange blue
Generate a plan, i.e., a sequence of actions separated by commas, taking the agent in the environment from the initial state to the goal state.
*Assistant:* `unstack blue orange, putdown blue, pickup orange, stack orange blue, end`
*User:* Given an instance of the Blocksworld domain as follows:
Block Objects: red, blue, orange, yellow
Initial State: clear red, clear yellow, handempty, on red blue, on yellow orange, ontable blue, ontable orange
Goal State: on orange red
Generate a plan, i.e., a sequence of actions separated by commas, taking the agent in the environment from the initial state to the goal state.
*Assistant:* `unstack yellow orange, putdown yellow, pickup orange, stack orange red, end`

Figure 4: Prompt template for the Blocksworld planning task, showing the system message that defines predicates and action restrictions, followed by example interactions.

## B  Answer Set Programming

Answer Set Programming (ASP) is a declarative paradigm for knowledge representation and reasoning, where problems are specified logically, and solutions (answer sets) are computed via specialized solvers. An ASP program consists of rules: normal rules of the form $h \leftarrow b_1, \cdots, b_n$ (which is read as "$h$ is true if all $b_i$ are true"), choice rules $l\{h_1; \cdots; h_m\}u \leftarrow b_1, \cdots, b_n$ (specifying that between $l$ and $u$ atoms from $h_1, \cdots, h_m$ must be true when the body holds), hard constraints of the form $\leftarrow b_1, \cdots, b_n$ (ruling out answer sets where the body is satisfied), and weak constraints $\sim b_1, \cdots, b_n.[w@l]$ (assigning weight $w$ at level $l$ to penalize solutions that satisfy the body of the constraints).

Given an ASP program $P$, its semantics is defined through its Herbrand Base $HB_P$ (the set of all possible ground atoms) and Herbrand Interpretations (subsets of $HB_P$). ASP solvers compute answer sets $AS(P)$, i.e., interpretations that satisfy all rules in $P$. We refer the reader to Lifschitz (2019) for more thorough details.

## C  Task Breakdown

Table 7: Overview of tasks and their characteristics

| Class | Task | Example | Constraint | $\rho(\cdot)$ |
|---|---|---|---|---|
| Synthetic Grammar Synthesis | $a^n b^n c^n$ | $n = 3$: $aaabbbccc$ | Equal count of $a$, $b$, $c$ | $n - n_w$ |
| | $a^m b^n c^m d^n$ | $m = 2, n = 3$: $aabbbccdddd$ | Paired counts $m \neq n$ | $(m+n)-(m_w+n_w)$ |
| | $w\|w$ | $w = ab$: $abab$ | Exact string replication | Distance to target counts |
| Combinatorial Reasoning | Sudoku-3×3 | [1,*,3],[*,*,2],[*,1,*] | Row, column constraints | 0 if solved, 1 otherwise |
| | Sudoku-4×4 | [1,*,3,4],[*,*,2,*],[*,1,*,2] | Row, column, box constraints | 0 if solved, 1 otherwise |
| | 3-Graph Coloring | $(2,1)(2,0)(0,2)$; Edges: $[0,1,2]$ | Valid 3-color assignment | $e - e_w$ |
| Parsing | JSON | {"item": "book", "price": 15} | JSON schema (CFG) | - (no reward) |
| Blocks Planning | Plan Gen. | pickup red, stack red blue, end | State validity + preconditions | $h(s,g)+ \alpha \cdot \text{len(plan)}$ |

Table 7 provides a comprehensive overview of all tasks evaluated in our experiments. Each row details a specific task, showing example inputs/outputs, constraints that must be satisfied, and distance functions used to evaluate solution quality/correctness. The tasks span four categories: Synthetic Grammar Synthesis (testing formal language generation with increasing complexity), Combinatorial Reasoning (assessing structured problem-solving requiring combinatorial reasoning), Parsing (demonstrating semi-open-ended structural generation capabilities where semantic constraints and correctness rewards are not applicable), and Blocks Planning (evaluating semantic constraints and sequential decision-making with state-dependent constraints).

# D  Answer Set Grammar Examples

```
start → board {} % This is an example of a purely context-free production rule

board → "[" row , row , row , row "]" {
    % This is a context-sensitive production rule with the following semantic constraints:
    % Assign each cell's value to a corresponding (X, Y) coordinate
    cell_value((1,1),V1) :- col(1,V1)@2.
    cell_value((1,2),V2) :- col(2,V2)@2.
    ⋮
    cell_value((4,4),V4) :- col(4,V4)@8.

    % Sudoku constraints: Same row, column, and block
    :- same_row(C1,C2), cell_value(C1,V), cell_value(C2,V).
    :- same_col(C1,C2), cell_value(C1,V), cell_value(C2,V).
    :- same_block(C1,C2), cell_value(C1,V), cell_value(C2,V).
}

row → "[" cell , cell , cell , cell "]" {
    col(1,V1) :- cell_value(V1)@2.
    ⋮
    col(4,V4) :- cell_value(V4)@8.
}

cell → digit { cell_value(X) :- digit(X)@1. }

digit → "1" { digit(1). } | "2" { digit(2). } | "3" { digit(3). } | "4" { digit(4). }

#background {
    % Cell coordinates
    cell((1,1)). cell((1,2)). … cell((4,4)).

    % Cell values
    cell_value((1, 1), 1). …

    % Block definitions
    block((X,Y),tl) :- X<3, Y<3.
    ⋮
    block((X,Y),br) :- X>2, Y>2.

    % Structural relations
    same_row((X1,Y),(X2,Y)) :- X1 != X2.
    same_col((X,Y1),(X,Y2)) :- Y1 != Y2.
    same_block(C1,C2) :- block(C1,B), block(C2,B), C1 != C2.
}
```

Figure 5: Example ASG for a 4×4 Sudoku puzzle. The context-free portion of the ASG is given by the grammar productions, while semantic constraints are expressed as bold ASP annotations within curly braces; omitting these annotations yields a standard CFG. For instance, the `start` production rule is context-free, as it carries no ASP annotations specifying semantic constraints. Answer Set Programming (ASP) annotations define grid structure, map parsed digits to grid coordinates, and enforce Sudoku constraints over rows, columns, and blocks.

```
start → action_seq {} % This is an example of a purely context-free production rule

action_seq → actions , action_seq {} | end {} | {}

actions → stack {} | unstack {} | pickup {} | putdown {}

pickup → "pickup" block {
    % This is a context-sensitive production rule with the following semantic constraints:
    :- block(X)@2, not handempty.
    :- block(X)@2, not clear(X)@2.
    :- block(X)@2, not ontable(X)@2.
    :- goal.
}

putdown → "putdown" block {
    :- block(X)@2, not holding(X)@2.
    :- goal.
}

stack → "stack" block block {
    :- block(X)@2, block(Y)@3, X = Y.
    :- block(X)@2, not holding(X)@2.
    :- block(X)@3, not clear(X)@3.
    :- goal.
}

unstack → "unstack" block block {
    :- block(X)@2, block(Y)@3, X = Y.
    :- block(X)@2, block(Y)@3, not on(X,Y).
    :- block(X)@2, not clear(X).
    :- block(X)@2, block(Y)@3, not handempty.
    :- goal.
}

end → "end" { :- not goal. }

comma → "," {}

block → "red" { block("red"). } | "blue" { block("blue"). } | "green" {
block("green"). }

#background {
    % Initial state
    ontable("red").
    on("blue","red").
    ontable("green").
    clear("blue").
    clear("green").
    handempty.
}
```

Figure 6: Example ASG for the Blocksworld domain. Grammar productions define a context-free backbone in standard Extended Backus–Naur form, while ASP annotations in bold and within curly braces specify semantic constraints; removing these annotations yields a CFG. Answer Set Programming (ASP) annotations enforce action preconditions, state consistency, and goal satisfaction over generated action sequences.

```
start → as bs cs {
% This is a context-sensitive production rule with the following semantic constraints:
      :- size(X)@1, not size(X)@2.
      :- size(X)@1, not size(X)@3.
}

as → "a" as { size(X+1) :- size(X)@2. }  |  { size(0). }

bs → "b" bs { size(X+1) :- size(X)@2. }  |  { size(0). }

cs → "c" cs { size(X+1) :- size(X)@2. }  |  { size(0). }
```

Figure 7: Example ASG for the language $a^n b^n c^n$, where context-sensitive equal-length constraints are introduced via ASP annotations in bold and enclosed in curly braces. If these annotations are removed (i.e., all curly braces are empty), the language defined by the grammar productions reduces to $a^i b^j c^k$.

```
% This ASG encodes a pure CFG without any ASP semantic constraints over
production rules in curly braces

start → "{" fields "}" {}

fields → firstName , lastName , age {}

firstName → "firstName" ":" string {}

lastName → "lastName" ":" string {}

age → "age" ":" number {}
```

Figure 8: Example ASG for a simplified fragment of JSON, where all semantic annotation blocks are empty, and the formalism reduces to a pure CFG written in standard Extended Backus–Naur form.

# E   Complete Baseline Methods

We systematically evaluate all combinations of sampling algorithms (Base, BoN, MCTS) with constraint types (unconstrained, $\mathcal{C}_{\text{CFG}}$, $\mathcal{C}_{\text{CSG}}$) to provide comprehensive ablation analysis. The complete list of baseline methods is as follows:

1. **Base Unconstrained:** Greedy sampling to assess the base model's capability.

2. **Base with $\mathcal{C}_{\textbf{CFG}}$:** Locally constrained syntactic decoding. This approach is analogous to various work in controlled decoding where the LLM's next tokens are masked according to CFG constraints (Geng et al., 2023; Ugare et al., 2024; Beurer-Kellner et al., 2024, interalia), though we implement this through an ASG encoding a CFG.

3. **XGrammar (Dong et al., 2024):** While Base with $\mathcal{C}_{\text{CFG}}$ is functionally equivalent to prior work in syntactic control, we run an additional baseline for the parsing task for completeness and to empirically highlight this equivalence.

4. **Base with $\mathcal{C}_{\textbf{CSG}}$:** We mask LLM logits with terminals from an ASG encoding context-sensitive and semantic constraints. This parallels work in semantic parsing, though prior methods typically use CFG with ad-hoc constraints (e.g., Scholak et al., 2021; Poesia et al., 2022; Roy et al., 2023).

5. **BoN Unconstrained:** Best-of-N (BoN) serves as a simple constraint satisfaction mechanism by sampling $N$ generations and rejecting invalid samples according to $\mathcal{C}$ and ranking solutions by $\mathcal{R}(\cdot)$ Welleck et al. (2024). We set $N$ to match `SEM-CTRL`'s computational budget (maximum number of MCTS samples generated during search) for fair comparison. See Appendix F for $N$ values.

6. **BoN with $\mathcal{C}_{\textbf{CFG}}$:** We apply rejection sampling with local syntactic constraints.

7. **BoN with $\mathcal{C}_{\textbf{CSG}}$:** This serves as an additional ablation against `SEM-CTRL` to ascertain whether `SEM-CTRL`'s search-guided reasoning capability and token-level incorporation of solution quality induces improvements. This is the first baseline that incorporates both notions of semantic validity and solution correctness.

8. **MCTS Unconstrained:** MCTS applied at the token level, corresponding to a range of search-guided reasoning approaches (e.g., Zhang et al., 2023b; Wan et al., 2024).

9. **MCTS with $\mathcal{C}_{\textbf{CFG}}$:** Here, we run MCTS with an ASG only encoding syntactic constraints to assess if the model benefits from additional semantic guidance and pruning achieved by `SEM-CTRL`.

10. **`SEM-CTRL`:** Our complete approach combining semantic constraints ($\mathcal{C}_{\text{CSG}}$) with MCTS for semantically guided search.

# F   Further LLM Sampling Parameters

Table 8: Maximum sample times ($N$) for BoN and MCTS/`SEM-CTRL`

| Task | Sample Times ($N$) |
|---|---|
| SGS | 50 |
| Graph Coloring | 35 |
| Sudoku 3×3 | 10 |
| Sudoku 4×4 | 265 |
| Blocks | 200 |

We have already provided most of the LLM sampling parameter details in Section 4.2. Here, we define the max sample times for BoN and `SEM-CTRL`/MCTS. We empirically select the maximum sample times ($N$) for both BoN and `SEM-CTRL`/MCTS based on task complexity, as shown in Table 8. In BoN, an LLM samples $N$ generations, then the reward function $\mathcal{R}(y_t)$ selects the best output according to the highest reward. In MCTS, we conduct the four MCTS steps (discussed in Section 3.3.2) until we sample $N$ full generations. Note: for the JSON parsing task, we only sample one generation for all methods (operate under Base sampling), given the lack of a reward function.

For computational feasibility, we make an exception for Llama 70B BoN in Blocks tasks: we use 15 samples (matching `SEM-CTRL`'s average) for main experiments and 50 samples (31 more than `SEM-CTRL`'s average) for ablation studies, rather than the full 200 samples, as generating this many sequences from a 70B model across 600 problems would be prohibitively expensive.

## G    Compute Cluster Specifications

Our experiments were conducted using two GPU clusters. The first cluster used nodes with $2\times$ Intel Xeon Platinum 8358 CPUs (2.60GHz, 32 cores each) and NVIDIA L40S GPUs (48GB GDDR6), where we utilized up to 4 GPUs with 1TB RAM per node. The second cluster used nodes with $2\times$ Intel Xeon Platinum 8360Y CPUs (2.40GHz, 36 cores each) and NVIDIA A100 GPUs (80GB), where we utilized up to 2 GPUs with 1TB RAM per node.

## H    Further Results

This section provides additional experimental results, including those referenced in the main text and further analyses that support our findings.

### H.1    Context-Free and Context-Sensitive Correctness Results on Combinatorial Reasoning and Planning

In Section 5, we discussed the abilities of LLMs to conform to CFG and CSG constraints without enforcing any form of control across various sampling strategies, contrasting such results with `SEM-CTRL` and its impact on accuracy. The discussion specifically presented results on Synthetic Grammar Synthesis in Table 2. Here, we present the same tables on the tasks Combinatorial Reasoning and Planning in Table 9 and Table 10, respectively. As detailed in Section 5, similar conclusions can be drawn according to such results. Hence, any further discussions are omitted.

Table 9: Accuracy (A), context-free ($V_{\mathrm{CFG}}$) and context-sensitive ($V_{\mathrm{CSG}}$) correctness results on Combinatorial Reasoning

| Alg. | Model | A | $V_{\mathrm{CFG}}$ | $V_{\mathrm{CSG}}$ |
|---|---|---|---|---|
| Base | Llama 1B | 0.0% | 11.0% | 4.0% |
| | Llama 8B | 25.0% | 79.0% | 32.0% |
| | Llama 70B | 57.4% | 96.0% | 66.0% |
| BoN | Llama 1B | 0.0% | 18.0% | 10.0% |
| | Llama 8B | 65.5% | 87.0% | 73.0% |
| | Llama 70B | 89.7% | 99.0% | 91.0% |
| API | o1-preview | 92.9% | 100.0% | 99.0% |
| | DeepSeek-R1 | 92.9% | 100.0% | 96.0% |
| | o4-mini | 92.9% | 100.0% | 100.0% |
| `SEM-CTRL` | Llama 1B | 100.0% | 100.0% | 100.0% |
| | Llama 8B | 100.0% | 100.0% | 100.0% |
| | Llama 70B | 100.0% | 100.0% | 100.0% |

Table 10: Accuracy (A), context-free ($V_{CFG}$) and context-sensitive ($V_{CSG}$) correctness results on Planning (Blocks)

| Alg. | Model | A | $V_{CFG}$ | $V_{CSG}$ |
|---|---|---|---|---|
| Base | Llama 1B | 0.0% | 100.0% | 0.0% |
| | Llama 8B | 2.0% | 100.0% | 2.8% |
| | Llama 70B | 23.2% | 100.0% | 28.7% |
| BoN | Llama 1B | 4.3% | 86.2% | 5.2% |
| | Llama 8B | 27.3% | 99.2% | 27.8% |
| | Llama 70B | 48.8% | 98.8% | 54.7% |
| API | o1-preview | 94.5% | 99.8% | 94.7% |
| | DeepSeek-r1 | 96.5% | 99.5% | 97.2% |
| | o4-mini | 98.5% | 99.0% | 98.5% |
| SEM-CTRL | Llama 1B | 74.0% | 100.0% | 100.0% |
| | Llama 8B | 90.3% | 100.0% | 100.0% |
| | Llama 70B | 96.8% | 100.0% | 100.0% |

## H.2 Extended Results with Llama 3.1 8B

The results presented in Section 4 showcased model performance on Llama 3.2 1B and Llama 3.1 70B. Here, we present the full results with Llama 3.1 8B, which was omitted from the original tables due to space requirements. While similar findings can be drawn, we chose to present models at opposite ends of the parameter scale (1B and 70B) in the main text. Table 11 corresponds to Table 1, and Table 12 corresponds to Table 2 in the main text. Similarly, Table 9 and Table 10 showcase the 8B results on Combinatorial Reasoning and Planning, respectively.

Table 11: Accuracy (A) results for all tasks: $a^n b^n c^n$, $a^m b^n c^m d^n$, Copy, Graph Coloring (Graph), Sudoku 3x3 (Sudoku-3), Sudoku 4x4 (Sudoku-4), and Blocksworld Planning (Blocks), using various sampling algorithms (Alg.), or API in the case of o1-preview, DeepSeek-R1, and o4-mini, with different base LLMs (Model). Here, we include Llama 3.1 8B.

| Alg. | Model | Synthetic Grammar Synthesis | | | Combinatorial Reasoning | | | Planning |
|---|---|---|---|---|---|---|---|---|
| | | $a^n b^n c^n$ | $a^m b^n c^m d^n$ | Copy | Graph | Sudoku-3 | Sudoku-4 | Blocks |
| Base | Llama 1B | 3.3% | 0.0% | 10.0% | 0.0% | 0.0% | 0.0% | 0.0% |
| | Llama 8B | 3.3% | 0.0% | 13.3% | 25.0% | 40.0% | 10.0% | 2.0% |
| | Llama 70B | 30.0% | 0.0% | 60.0% | 37.5% | 90.0% | 30.0% | 23.2% |
| BoN | Llama 1B | 7.8% | 1.1% | 48.9% | 0.0% | 0.0% | 0.0% | 4.3% |
| | Llama 8B | 26.7% | 13.3% | 98.9% | 41.7% | 70.0% | 80.0% | 27.3% |
| | Llama 70B | 71.1% | 22.2% | 88.9% | 100.0% | 96.7% | 80.0% | 48.8% |
| API | o1-preview | 83.3% | 80.0% | 96.7% | 75.0% | 100.0% | 100.0% | 94.5% |
| | DeepSeek-R1 | 83.3% | 70.0% | 96.7% | 75.0% | 100.0% | 100.0% | 96.5% |
| | o4-mini | 93.3% | 93.3% | 100.0% | 75.0% | 100.0% | 100.0% | 98.5% |
| SEM-CTRL | Llama 1B | 100.0% | 100.0% | 100.0% | 100.0% | 100.0% | 100.0% | 74.0% |
| | Llama 8B | 100.0% | 100.0% | 100.0% | 100.0% | 100.0% | 100.0% | 90.3% |
| | Llama 70B | 100.0% | 100.0% | 100.0% | 100.0% | 100.0% | 100.0% | 96.8% |

Table 12: Synthetic Grammar Synthesis tasks results (with Llama 8B)

| Alg. | Model | A | $V_{CFG}$ | $V_{CSG}$ |
|------|-------|---|-----------|-----------|
| Base | Llama 1B | 4.4% | 79.0% | 23.0% |
|      | Llama 8B | 5.6% | 94.0% | 24.0% |
|      | Llama 70B | 30.0% | 99.0% | 39.0% |
| BoN | Llama 1B | 19.3% | 65.0% | 35.0% |
|     | Llama 8B | 46.3% | 91.0% | 59.0% |
|     | Llama 70B | 60.7% | 97.0% | 79.0% |
| API | o1-preview | 86.7% | 100.0% | 88.0% |
|     | DeepSeek-R1 | 83.3% | 100.0% | 87.0% |
|     | o4-mini | 94.7% | 99.0% | 95.0% |
| SEM-CTRL | Llama 1B | 100.0% | 100.0% | 100.0% |
|          | Llama 8B | 100.0% | 100.0% | 100.0% |
|          | Llama 70B | 100.0% | 100.0% | 100.0% |

## H.3 Soft Accuracies

In Table 1, we demonstrate the performance of all models and baselines according to binary accuracy (1 if solved, 0 otherwise). Here, we also introduce a complementary metric, **Soft Accuracy**, which uses the raw values from the reward function defined in Equation (6). Unlike binary accuracy, soft accuracy shows a continuous progression of results, where partial solutions receive credit proportional to their proximity to the correct solution, as determined by $\rho(\cdot)$. Table 13 presents these results, which provide additional nuance to our model comparisons.

Table 13: Soft Accuracy (%) results for all tasks

| | | Synthetic Grammar Synthesis | | | Combinatorial Reasoning | | |
|------|-------|-------------|-------------|------|-------|----------|----------|
| Alg. | Model | $a^n b^n c^n$ | $a^m b^n c^m d^n$ | Copy | Graph | Sudoku-3 | Sudoku-4 |
| Base | Llama 1B | 6.1±20.5% | 2.1±11.3% | 40.5±40.5% | 10.4±28.1% | 0.0±0.0% | 0.0±0.0% |
|      | Llama 8B | 7.7±24.1% | 0.0±0.0% | 42.5±42.4% | 25.0±44.2% | 40.0±49.8% | 10.0±30.5% |
|      | Llama 70B | 30.0±46.1% | 3.1±16.6% | 77.4±37.2% | 37.5±51.8% | 90.0±30.5% | 30.0±46.6% |
| BoN | Llama 1B | 16.0±33.0% | 5.4±20.8% | 67.1±41.1% | 24.3±36.0% | 0.0±0.0% | 0.0±0.0% |
|     | Llama 8B | 39.3±47.6% | 33.7±44.0% | 98.9±10.5% | 52.4±49.5% | 70.0±46.6% | 80.0±40.7% |
|     | Llama 70B | 78.3±40.8% | 59.6±45.3% | 91.1±26.6% | 100.0±0.0% | 96.7±18.3% | 80.0±40.7% |
| API | o1-preview | 83.3±37.5% | 80.6±39.4% | 98.6±10.7% | 92.0±21.3% | 100.0±0.0% | 100.0±0.0% |
|     | DeepSeek-R1 | 87.6±31.6% | 70.0±46.6% | 99.6±2.3% | 85.0±35.0% | 100.0±0.0% | 100.0±0.0% |
|     | o4-mini | 93.3±25.1% | 93.3±25.4% | 100.0±0.0% | 96.1±7.6% | 100.0±0.0% | 100.0±0.0% |
| SEM-CTRL | Llama 1B | 100.0±0.0% | 100.0±0.0% | 100.0±0.0% | 100.0±0.0% | 100.0±0.0% | 100.0±0.0% |
|          | Llama 8B | 100.0±0.0% | 100.0±0.0% | 100.0±0.0% | 100.0±0.0% | 100.0±0.0% | 100.0±0.0% |
|          | Llama 70B | 100.0±0.0% | 100.0±0.0% | 100.0±0.0% | 100.0±0.0% | 100.0±0.0% | 100.0±0.0% |

From these soft accuracy results, we make several key observations:

1. The high variance in base models and BoN (shown by large standard deviations) suggests inconsistent performance even when measuring partial correctness, highlighting the inherent unreliability of unconstrained approaches.

2. The progression from Llama 1B to 70B shows more gradual improvement under soft accuracy compared to binary accuracy, indicating that larger models not only solve more problems but also get 'closer' to correct solutions when they fail.

3. Tasks like Copy and Sudoku-3×3 show higher soft accuracy than binary accuracy across all baselines, suggesting these tasks may be easier to partially solve but challenging to get exactly right. In contrast, $a^m b^n c^m d^n$ shows similar scores in both metrics, indicating an 'all-or-nothing' task structure.

4. `SEM-CTRL` maintains perfect scores with zero variance across all models and tasks under both metrics, further validating its reliability regardless of the evaluation criteria.

### H.4 More Ablation Studies

In Section 5, we conduct an ablation study on the PlanBench benchmark Valmeekam et al. (2024), comparing our approach against prior work's sampling algorithms like BoN and Greedy with $\mathcal{C}_{\text{CFG}}$. Here, we extend this analysis to Synthetic Grammar Synthesis and Combinatorial Reasoning, with results shown in Tables 14 and 15. While our findings largely mirror those from the Blocksworld domain, we observe an interesting phenomenon with Llama 8B: BoN with $\mathcal{C}_{\text{CFG}}$ performs worse than its unconstrained version. This aligns with prior work (e.g., Park et al., 2024) showing that local decoding can misalign the LLM's generative distribution by naively redistributing probability mass from invalid tokens, inflating the likelihood of rare sequences. `SEM-CTRL` avoids this issue through its combination of semantic control and principled search guided by domain-specific rewards.

Table 14: Synthetic Grammar Synthesis ablation tasks results

| Alg. | Model | $\mathcal{C}$ | A | $\mathbf{V_{CFG}}$ | $\mathbf{V_{CSG}}$ |
|---|---|---|---|---|---|
| Base | Llama 1B | - | 4.4% | 79.0% | 23.0% |
| | Llama 1B | $\mathcal{C}_{\text{CFG}}$ | 4.4% | 100.0% | 23.0% |
| | Llama 1B | $\mathcal{C}_{\text{SEM}}$ | 11.1% | 100.0% | 100.0% |
| | Llama 8B | - | 5.6% | 94.0% | 24.0% |
| | Llama 8B | $\mathcal{C}_{\text{CFG}}$ | 8.9% | 100.0% | 34.0% |
| | Llama 8B | $\mathcal{C}_{\text{SEM}}$ | 21.1% | 100.0% | 100.0% |
| | Llama 70B | - | 30.0% | 99.0% | 39.0% |
| BoN | Llama 1B | - | 19.3% | 65.0% | 35.0% |
| | Llama 1B | $\mathcal{C}_{\text{CFG}}$ | 26.7% | 100.0% | 50.0% |
| | Llama 1B | $\mathcal{C}_{\text{SEM}}$ | 71.5% | 100.0% | 100.0% |
| | Llama 8B | - | 46.3% | 91.0% | 59.0% |
| | Llama 8B | $\mathcal{C}_{\text{CFG}}$ | 39.6% | 100.0% | 60.0% |
| | Llama 8B | $\mathcal{C}_{\text{SEM}}$ | 85.9% | 100.0% | 100.0% |
| | Llama 70B | - | 60.7% | 97.0% | 79.0% |
| MCTS | Llama 1B | - | 20.0% | 80.0% | 33.0% |
| | Llama 1B | $\mathcal{C}_{\text{CFG}}$ | 27.8% | 100.0% | 70.0% |
| | Llama 8B | - | 36.7% | 97.0% | 47.0% |
| | Llama 8B | $\mathcal{C}_{\text{CFG}}$ | 36.7% | 100.0% | 78.0% |
| API | o1-preview | - | 86.7% | 100.0% | 88.0% |
| | DeepSeek-R1 | - | 83.3% | 100.0% | 87.0% |
| | o4-mini | - | 94.7% | 99.0% | 95.0% |
| `SEM-CTRL` | Llama 1B | $\mathcal{C}_{\text{SEM}}$ | 100.0% | 100.0% | 100.0% |
| | Llama 8B | $\mathcal{C}_{\text{SEM}}$ | 100.0% | 100.0% | 100.0% |
| | Llama 70B | $\mathcal{C}_{\text{SEM}}$ | 100.0% | 100.0% | 100.0% |

Table 15: Combinatorial Reasoning ablation tasks results

| Alg. | Model | $\mathcal{C}$ | A | $\mathbf{V_{CFG}}$ | $\mathbf{V_{CSG}}$ |
|---|---|---|---|---|---|
| Base | Llama 1B | - | 0.0% | 11.0% | 4.0% |
| | Llama 1B | $\mathcal{C}_{\text{CFG}}$ | 3.6% | 100.0% | 25.0% |
| | Llama 1B | $\mathcal{C}_{\text{SEM}}$ | 14.3% | 100.0% | 100.0% |
| | Llama 8B | - | 25.0% | 79.0% | 32.0% |
| | Llama 8B | $\mathcal{C}_{\text{CFG}}$ | 21.1% | 100.0% | 32.0% |
| | Llama 8B | $\mathcal{C}_{\text{SEM}}$ | 28.6% | 100.0% | 100.0% |
| | Llama 70B | - | 57.4% | 96.0% | 66.0% |
| BoN | Llama 1B | - | 0.0% | 18.0% | 10.0% |
| | Llama 1B | $\mathcal{C}_{\text{CFG}}$ | 19.1% | 100.0% | 39.0% |
| | Llama 1B | $\mathcal{C}_{\text{SEM}}$ | 76.2% | 100.0% | 100.0% |
| | Llama 8B | - | 65.5% | 87.0% | 73.0% |
| | Llama 8B | $\mathcal{C}_{\text{CFG}}$ | 77.4% | 100.0% | 81.0% |
| | Llama 8B | $\mathcal{C}_{\text{SEM}}$ | 94.0% | 100.0% | 100.0% |
| | Llama 70B | - | 89.7% | 99.0% | 91.0% |
| MCTS | Llama 1B | - | 0.0% | 18.0% | 14.0% |
| | Llama 1B | $\mathcal{C}_{\text{CFG}}$ | 32.1% | 100.0% | 61.0% |
| | Llama 8B | - | 82.1% | 93.0% | 82.0% |
| | Llama 8B | $\mathcal{C}_{\text{CFG}}$ | 85.7% | 100.0% | 93.0% |
| API | o1-preview | - | 92.9% | 100.0% | 99.0% |
| | DeepSeek-R1 | - | 92.9% | 100.0% | 96.0% |
| | o4-mini | - | 92.9% | 100.0% | 100.0% |
| SEM-CTRL | Llama 1B | $\mathcal{C}_{\text{SEM}}$ | 100.0% | 100.0% | 100.0% |
| | Llama 8B | $\mathcal{C}_{\text{SEM}}$ | 100.0% | 100.0% | 100.0% |
| | Llama 70B | $\mathcal{C}_{\text{SEM}}$ | 100.0% | 100.0% | 100.0% |

