# OpenReview forum: "$\texttt{SEM-CTRL}$: Semantically Controlled Decoding"
_TMLR — Accepted by TMLR_

### Review · Reviewer_aSC2 · 2025-10-22

**Summary Of Contributions:**

The authors propose SEM-CTRL, a framework which uses MCTS methods on constrained LLM outputs to improve performance on reasoning tasks. The core idea is this: once an LLM's output is correctly constrained to a grammar (in this case, an "answer set grammar", which defines "task-specific semantics"), it becomes possible to model and analyze sequences of decisions/outputs to better inform the decoding of the generated answer. The authors show that this principle, combined with their MCTS approach, improves the performance of off-the-shelf language models on a variety of benchmarks, allowing for small LLMs to achieve performance that normally requires more expensive models to achieve with previous methods.

**Audience:**

Yes

**Audience Explanation:**

This work seems obviously relevant for audiences interested in ML/AI, since it focuses on popular language models and their applications.

**Broader Impact Concerns:**

I have no broader impact concerns that were not addressed in the provided statement in the paper already.

**Claims And Evidence:**

Yes

**Claims Explanation:**

The authors took care to benchmark their approach on a variety of datasets and across different popular LLM models, and the results are consistent with their claims that SEM-CTRL boosts performance, even with small models (such as llama-1b). The effect of the SEM-CTRL approach is studied explicitly via ablation, which shows improvement over a more simplistic MCTS approach that does not use the grammar-based approach, and constrained sampling approaches that do not use MCTS. The discussion at the end about broader impact and limitations of the work is appreciated.

**Requested Changes:**

The work discusses how the proposed approach is related to "global correctness optimization". As someone mostly unfamiliar with this field, it would be nice to see more explanation of other approaches, and possibly comparison.

One of the advertised benefits is the ability to achieve strong performance while avoiding fine tuning. But the work focuses on SEM-CTRL's inference-time performance. The high cost of fine tuning happens at training time: once a model is deployed for inference there is no difference (assuming same model size). Thus, if one is interested in just inference performance, it makes sense to compare any claims against results with fine tuning. Is it worth using a non-fine-tuned model of the same size equipped with SEM-CTRL? What benefits are there to using SEM-CTRL together with fine tuning? Or, if aiming for a certain level of performance at a certain model size (above what is possible with previous decoding approaches on a non-fine-tuned model), does using SEM-CTRL reduce the amount of fine tuning needed to reach the target? I would be curious to see further analysis of this.

---

> ### Author Response · Authors · 2025-10-28
> **Response to reviewer aSC2**
>
> We thank you for thorough feedback and comments, and for appreciating our benchmarking efforts and the relevance of $\texttt{SEM-CTRL}$ to the broader AI community. We appreciate the opportunity to clarify some of the questions and indeed believe that addressing them will further strengthen the paper. In the following parts, we address the salient questions below:
>
> ---
>
> ### On Global Correctness Optimization
>
>
> We will revise the Introduction and Analysis sections to make this argument more explicit in the draft. Our focus on correctness in $\texttt{SEM-CTRL}$ lies along two dimensions:
>
> **(1) Validity:** whether the generated output is syntactically and semantically valid (as is captured by an Answer Set Grammar (ASG)). $\texttt{SEM-CTRL}$ guarantees 100% validity since all token expansions have to satisfy ASG rules (we explain this further in Sec. 3.2.1). For e.g., consider a Blocksworld planning domain where an agent can pick up or put down blocks. A sequence such as $\texttt{pickup(A), putdown(A), pickup(B)}$ is valid because it respects the grammar and action preconditions, even if it does not solve the goal. ASGs capture this notion of validity.
>
> **(2) Task‑level correctness:** whether the final output successfully solves the downstream task. Here, we explore MCTS to explore the space of valid outputs and globally optimizes token sequences under the task reward. Thus, “global correctness optimization” refers to this MCTS‑driven search constrained by ASG validity. If a correct solution exists at some depth _d_, $\texttt{SEM-CTRL}$ is guaranteed to find it given sufficient search time; under limited budgets, it optimizes toward the best reachable solution. For e.g., continuing the previous example, $\texttt{pickup(A), putdown(A), pickup(B)}$ is not correct if the goal is to stack A on B, while $\texttt{pickup(A), stack(A,B)}$ is correct. MCTS captures this second notion of task-level correctness.
>
> We hope our exposition above and examples have further clarified our framing of global correctness optimisation.
>
> In the Introduction (Section 1) and Section 3.3, we also illustrate the distinction between validity and correctness.
>
> Furthermore, all baselines we include, e.g., Best‑of‑N (BoN) rejection sampling, unconstrained MCTS, and reasoning models (e.g., o1‑preview, DeepSeek‑R1), are themselves correctness‑optimizing methods, as they explicitly seek high‑reward outputs at inference time. BoN, performs global optimization by sampling and ranking complete sequences (see [Welleck, S., et al (2024)]); unconstrained MCTS is a standard correctness‑optimization approach using principled search (i.e., [Zhang, S., et al (2023), Hao, S. et al (2023), Wan et al. (2024)]); reasoning models use internal reasoning trajectories to yield better generations. However, these lack explicit validity guarantees, leading to invalid or inconsistent generations. $\texttt{SEM-CTRL}$ unifies global optimization with guaranteed semantic validity and consistently outperforms these methods as shown in our ablations (Sec. 5.3). Furthermore, our framework applies correctness optimization under progressively stronger constraints (from syntactic to semantic), showing that richer grammars yield increasing gains in both validity and correctness. We contextualize these works in the literature and discuss similar/other approaches in Sec. 6 (“LLMs-Based Reasoning, Search, and Planning”).

---

> ### Author Response · Authors · 2025-10-28
> **Response to reviewer aSC2 (Cont.)**
>
> ### Fine‑tuning vs. $\texttt{SEM-CTRL}$
>
> We agree that a comparison to a fine-tuned model provides a valuable upper-bound reference. Indeed, our focus throughout the paper was to emphasize that $\texttt{SEM-CTRL}$ is primarily an inference‑time algorithm, designed to enable strong task performance and constraint satisfaction without fine‑tuning. However, considering fine-tuning is an excellent suggestion. We hypothesize that $\texttt{SEM-CTRL}$ could indeed be complementary to fine-tuning, perhaps to *steer* an already fine-tuned model towards higher-correctness outputs or to mitigate common failure modes (e.g., hallucination) that fine-tuning might not fully resolve.
>
> To further validate this claim empirically, we conducted additional experiments on the Blocksworld task, fine‑tuning Llama‑1B models with varying amounts of data from the Planbench dataset (5%, 10%, 20%, 50%, and 100%) and comparing both greedy decoding and $\texttt{SEM-CTRL}$. We report two key metrics:
>
> 1. **Accuracy:** percentage of problems solved
> 2. **Sample Efficiency:** the number of full sampled sequences MCTS explores to find a correct solution
> Results are evaluated on the ablation set (Sec. 5.3).
>
> | Model                     | Fine‑tuning Data | Problems Solved (%) | Avg. Sampled Sequences |
> |---------------------------|------------------|---------------------|------------------------|
> | **Greedy (Llama‑1B)**     | 0%             | 0                   | –                      |
> |                           | 5 %              | 12                  | –                      |
> |                           | 10 %             | 10                  | –                      |
> |                           | 20 %             | 16                  | –                      |
> |                           | 50 %             | 16                  | –                      |
> |                           | Full | 14                  | –                      |
> | **$\texttt{SEM-CTRL}$ (Llama‑1B)** | 0%             | 76                  | 71.14                  |
> |                           | 5 %              | 58                  | 53.80                  |
> |                           | 10 %             | 76                  | 46.00                  |
> |                           | 20 %             | 90                  | 30.04                  |
> |                           | 50 %             | 88                  | 17.44                  |
> |                           | Full  | 88                  | **13.22**              |
> | **$\texttt{SEM-CTRL}$ (Llama‑70B)**| 0%             | **98**              | **19.12**              |
>
> **Our key empirical observations are the following:**
>
> - Without fine‑tuning, $\texttt{SEM-CTRL}$ (Llama‑1B) already outperforms fine‑tuned greedy baselines by a large margin (76 % vs. 16 %), confirming that $\texttt{SEM-CTRL}$ alone provides strong inference‑time gains.
> - Fine‑tuning and $\texttt{SEM-CTRL}$ are **complementary**, we see that combining both indeed improves both accuracy (up to +14%) and sample efficiency (up to 5.4× fewer sampled sequences).
> - Sample efficiency improves as fine‑tuning data increases.
> - Compared to Llama‑70B, the 1B + $\texttt{SEM-CTRL}$ model (88‑90 %) approaches 70B + $\texttt{SEM-CTRL}$ (98 %), despite being orders of magnitude smaller and up to 1.4× more sample efficient. Thus, validating $\texttt{SEM-CTRL}$’s parameter efficiency.
>
> These match with the initial hypothesis. We will add this as an additional section in the new draft of the paper.
>
> We thank the reviewer once again for their thoughtful comments and feedback, and hope our clarifications and additional experiments have addressed the concerns. We will incorporate all these improvements and make it more explicit in the new draft.
>
> ### References
> - Welleck, S., et al, TMLR, 2024, https://arxiv.org/abs/2406.16838.
> - Zhang, S., et al. ICLR, 2023, https://openreview.net/forum?id=Lr8cOOtYbfL.
> - Hao, S. et al, EMNLP, 2023, https://arxiv.org/abs/2305.14992.
> - Wan et al., ICML, 2024, https://proceedings.mlr.press/v235/wan24c.html.

---

### Review · Reviewer_TZXh · 2025-12-04

**Summary Of Contributions:**

This paper introduces SEM-CTRL, a controlled decoding method that combines ASG-based semantic validity checking with MCTS-based search for solution correctness. Unlike methods that rely only on syntactic constraints or unconstrained MCTS, SEM-CTRL jointly enforces syntactic validity, semantic validity, and answer correctness. ASGs filter next-token candidates to those that can still form a semantically valid completion, while MCTS searches globally for high-reward (i.e., correct) solutions; all explored rollouts are semantically valid by construction.

SEM-CTRL is evaluated on synthetic formal languages, combinatorial reasoning tasks (Sudoku, graph coloring), JSON parsing, and Blocksworld planning. Across these benchmarks, SEM-CTRL achieves high accuracy (Table 1) and perfect grammatical and semantic validity (Tables 2 and 4). Notably, it also improves parameter efficiency: even Llama 1B matches or surpasses state-of-the-art reasoning models (o1-preview, o4-mini, DeepSeek-R1).

**Strengths**
- A novel integration of symbolic validity checking (via ASGs) with reward-guided search (via MCTS).
- Strong empirical results demonstrating high accuracy across diverse benchmarks.
- Provides semantic validity guarantees that even state-of-the-art reasoning-oriented LLMs cannot offer.
- Covers a broad range of tasks, including synthetic grammars, combinatorial puzzles, JSON parsing, and planning.
- Offers a rich set of experiments that go beyond accuracy, including grammatical validity analyses, ablation studies, and computational efficiency measurements.
- Includes (albeit brief) discussion of methods to reduce computational overhead (Section 3.4).

**Weaknesses**
- Authoring ASGs may require substantial expert knowledge and may not be feasible for more open-ended or poorly formalized domains. Relatedly, all evaluated tasks are synthetic or highly structured, leaving open the question of whether SEM-CTRL can be applied in more unconstrained, real-world settings.
- Semantic constraint checking can incur significant computational overhead (e.g., >100 seconds in SGS tasks). The absence of wall-clock execution times in Section 5.4 makes it difficult to fully understand or contextualize this overhead relative to end-to-end performance.

**Audience:**

Yes

**Audience Explanation:**

The paper is highly relevant to researchers working on controllable generation, structured prediction, logical reasoning with LLMs, and safe or verifiable model outputs. SEM-CTRL is likely to appeal to the TMLR audience because it offers a unique approach that simultaneously addresses controllable generation (semantic validity) and logical reasoning (task accuracy). The work is particularly notable in showing that a principled decoding method can outperform much larger reasoning-focused models, making it both timely and impactful.

**Broader Impact Concerns:**

The paper includes a Broader Impact Statement, which largely addresses environmental benefits and risks of bias encoded in constraints.

Additionally, below concerns could be addressed to enrich the Broader Impact Statement:
1. The statement could more explicitly discuss potential misuse cases. While guaranteed correctness is beneficial, it could also be used to generate formally valid but malicious content (e.g., syntactically correct exploits, valid but misleading formal arguments).
2. The democratization benefit of enabling smaller models is mentioned, but the requirement for ASG specification expertise may actually create new barriers to access. This tension could be acknowledged.
3. The paper should briefly mention that perfect syntactic/semantic validity doesn't guarantee truthfulness or ethical appropriateness of generated content.

These are relatively minor concerns, and the existing Broader Impact Statement demonstrates appropriate consideration of ethical implications. The suggested additions would strengthen it but are not critical omissions.

**Claims And Evidence:**

Yes

**Claims Explanation:**

The claims made in the submission are supported by convincing evidence.

Claim 1: (Performance enhancement) SEM-CTRL improves task performance.
 → This is supported by the near-perfect accuracy of SEM-CTRL–enabled models shown in Table 1.

Claim 2: (Parameter efficiency) SEM-CTRL enables small models to match or exceed the performance of much larger reasoning-oriented models.
 → This is demonstrated by comparing Llama 1B + SEM-CTRL against 70B unconstrained models and state-of-the-art reasoning models in Table 1.

Claim 3: (Validity guarantees) SEM-CTRL enforces grammatical and semantic correctness of outputs.
 → This is substantiated by the V_CFG, V_CSG, and V_SEM results reported in Tables 2 and 3.
While the computational overhead of semantic constraint checking is somewhat under-discussed, this does not undermine the core claims. Overall, the empirical evidence is strong and clearly presented.

**Requested Changes:**

**Critical changes (important for acceptance):**
1. **Discuss scalability limitations more transparently.**
 The ASG constraint-checking overhead is substantial in some benchmarks. A deeper analysis of computational complexity, memory usage, and scalability to longer sequences or larger grammars would strengthen the practical claims.
2. **Provide a more complete computational cost analysis.**
 Include wall-clock time comparisons (not just token counts) for SEM-CTRL versus baselines on the same hardware. Break down time spent on LLM inference, constraint checking, and search. This information is essential for practitioners to assess feasibility.
3. **Clarify the challenges and limitations of authoring ASGs for real-world domains.**
 The method assumes access to well-specified semantic rules. Discussion or concrete examples of how practitioners might construct ASGs in new or less formalized domains would help evaluate applicability and usability. Moreover, it would be helpful if the paper clarified whether ASGs can express all types of logic or are limited to particular families of logics, and where they can or cannot be applied. Explaining these limitations would make the scope of the method clearer.

**Non-critical suggestions (would improve the work but are not required):**
1. **Explore generalization beyond highly formal tasks.**
 Although the paper acknowledges limitations with open-ended tasks, consider adding an experiment demonstrating partial applicability to semi-structured generation (beyond JSON parsing), such as constrained story generation or semantic parsing with softer constraints.

---

> ### Author Response · Authors · 2025-12-15
> **Response to Reviewer TZXh (Part 1)**
>
> Thank you for the thoughtful comments and the feedback. We sincerely appreciate your recognition of the novelty of our approach, the parameter efficiency gains, and the strong empirical results across our experiments. We are also delighted that you found our work relevant to the broader research community’s goal of ensuring correct and verifiable model outputs for controllable generation and logical reasoning. We have organized our response across several replies for your convenience.
>
> ---
>
> > Discuss scalability limitations more transparently. The ASG constraint-checking overhead is substantial in some benchmarks. A deeper analysis of computational complexity, memory usage, and scalability to longer sequences or larger grammars would strengthen the practical claims.
>
> Thank you for the suggestion. We agree that elaborating on transparency will be useful. We will address this in greater detail in the revised draft. Here, we briefly address the concerns regarding theoretical computational complexity, scalability, and overheads as follows:
>
> Our work, $\texttt{SEM-CTRL}$, builds upon Answer Set Grammars (ASGs) introduced in Law, M., et al (2019). Law, M., et al (2019) have presented and proved the theoretical aspects, such as the complexity classes and the scalability of the ASG solver.
>
> Specifically, for the grammars introduced in our paper, the complexity class for constraint checks is exponential in the worst case. While we will briefly expand on this in the paper, we, however, would encourage the readers to refer to Law, M., et al (2019) for more details and proofs.

---

> ### Author Response · Authors · 2025-12-15
> **Response to Reviewer TZXh (Part 2)**
>
> > Provide a more complete computational cost analysis. Include wall-clock time comparisons (not just token counts) for SEM-CTRL versus baselines on the same hardware. Break down time spent on LLM inference, constraint checking, and search. This information is essential for practitioners to assess feasibility.
>
> We thank the reviewer for this extremely helpful suggestion.
>
> As an immediate response, we would like to note that the computational cost of $\texttt{SEM-CTRL}$ is dominated entirely by  (i) LLM forward passes (reported through total tokens generated per sample) and (ii) ASG constraint time (time spent on ASGs per sample). These two quantities, along with the MCTS search operations, constitute all of $\texttt{SEM-CTRL}$'s computational cost. We also note that the MCTS search operations, i.e., computing UCT scores, determining rollout/expansion decisions, maintaining visit counts, etc., are constant-time CPU operations (i.e., O(1)) that do not scale with model size, search depth, or number of tokens. Their contribution is negligible compared to the dominant GPU-bound LLM inference time and ASG constraint time.
>
> Formally, the total wall-clock cost decomposes as:
> \begin{equation}
> T_{total} \approx (N_{tokens} \times T_{LLM}) + T_{constraints} + T_{search\\_ops}
> \tag{1}
> \end{equation}
> where:
> - $N_{tokens}$ (Reported): The total number of tokens generated. This serves as the hardware-independent proxy for inference cost.
> - $T_{constraints}$ (Reported): Time spent on evaluating constraints.
> - $T_{LLM}$: The hardware- and framework-dependent latency of a single forward pass.
> - $T_{search\\_ops}$: Computationally instantaneous and constant time; therefore, it can be dropped from the equation.
>
> In our paper, we do not report $T_{LLM}$ but only $N_{tokens}$ as we followed previous work in token-level search, which only reports the number of generated tokens (e.g., [Valmeekam, K., et al (2025), Zhou, A., et al (2024), Wan, Z., et al (2024)]). Also, as our key baselines in the paper are closed-source API models running on unknown hardware with unknown inference strategies, direct wall-clock times would not be directly comparable across baselines. Instead, we consider the number of tokens to be a good proxy across models and methods (e.g., inference engines, model providers).
>
> In the revision, we will expand this discussion and incorporate the formal cost decomposition into Section 5.4 to highlight these points, thereby enabling practitioners to estimate wall-clock feasibility on their own hardware reliably.

---

> ### Author Response · Authors · 2025-12-15
> **Response to Reviewer TZXh (Part 3)**
>
> > Clarify the challenges and limitations of authoring ASGs for real-world domains. The method assumes access to well-specified semantic rules. Discussion or concrete examples of how practitioners might construct ASGs in new or less formalized domains would help evaluate applicability and usability.
>
> We again thank the reviewer for highlighting the importance of discussing the practical challenges of authoring ASGs and its expressiveness. We will discuss and incorporate this into the final manuscript to equip practitioners with the necessary context to adopt $\texttt{SEM-CTRL}$ for their own use cases. Here, we present a brief summary:
>
> An ASG consists of two distinct components: a standard CFG and constraints expressed in Answer Set Programming (ASP):
>
> 1) **Syntax (CFG)**: The CFG portion of an ASG is expressed in standard Extended Backus–Naur Form (EBNF) format. For instance, the JSON grammar used in Section 5.5 was automatically generated from Lark’s (a widely adopted Python parsing library) JSON CFG. Since CFGs for many tasks are widely available, we believe this step is accessible to general practitioners and relies on widely accessible tooling.
>
> To illustrate this, consider the language $a^nb^nc^n$, where the counts of all terminals must be equal in the specified order. Such a language can only be captured by a Context-Sensitive Grammar (CSG). A CFG instead can only capture the language $a^ib^jc^k$, enforcing only ordering without counting constraints. In ASGs, the CFG would be represented in standard EBNF format without any modifications as follows:
>
> ```
> start → as bs cs
> as → "a" as
> bs → "b" bs
> cs → "c" cs
> ```
>
> 2) **Semantics and Constraints (ASP):** Domain knowledge and production-rule constraints are expressed using ASP, a declarative logic programming paradigm.
>
> To continue with our previous example, in ASGs, to capture the context-sensitive counting constraints of $a^nb^nc^n$, ASP annotations over the production rules are added as follows:
>
> ```
> start → as bs cs {
>   :- size(X)@1, not size(X)@2.
>   :- size(X)@1, not size(X)@3.
> }
> as → "a" as { size(X+1) :- size(X)@2. } | { size(0). }
> bs → "b" bs { size(X+1) :- size(X)@2. } | { size(0). }
> cs → "c" cs { size(X+1) :- size(X)@2. } | { size(0). }
> ```
>
> As shown and as discussed in Section 2, this ASG extends its equivalent CFG by adding production rule constraints in curly braces ($\\{\dots\\}$) expressed via declarative ASP code.  The predicate $\texttt{size(X)}$ tracks the length $n$ of each sequence, where $\texttt{@i}$ indicates the i-th child position in the production rule. The constraint $\texttt{:- size(X)@1, not size(X)@2}$. reads as: “if the first child has size $\texttt{X}$, but the second child does not have size $\texttt{X}$, then reject this parse”. The production rules (except for the first one) contain two alternatives separated by ‘|’: the first alternative outputs one terminal symbol (i.e., $\texttt{a}$) and increments the size counter by one, using the value propagated from the recursively generated second child (i.e., $\texttt{size(X)@2}$), via the rule $\texttt{size(X+1) :- size(X)@2}$. The second branch initializes the base case counter with $\texttt{size(0)}$. Finally, the parse is accepted only if the ASP solver determines that the $\texttt{size}$ values propagated to the root of the parse tree satisfy the equality constraints, thereby guaranteeing the entire string conforms to the $a^nb^nc^n$ structure.
>
> We recognise that ASG development for very complex domains might require expertise of the ASP formalism. We believe that this can be circumvented by using LLMs to translate natural language constraints into the logical formalization of ASGs. For example, there is a line of recent work exploring this (e.g., [Alviano, M., et al (2025), Ishay, A., et al (2023)]).
>
> ---
>
> > Moreover, it would be helpful if the paper clarified whether ASGs can express all types of logic or are limited to particular families of logics, and where they can or cannot be applied. Explaining these limitations would make the scope of the method clearer.
>
> ASGs support the expressivity of ASP, which is a state-of-the-art, efficient computational first-order logic paradigm. As such, it can capture first-order logic with default reasoning and preference reasoning. But it does not capture logics such as fuzzy logic.
>
> In the revision, we will discuss and further expand on these points regarding the types of logic ASGs can capture and on the process of authoring new ASGs.

---

> ### Author Response · Authors · 2025-12-15
> **Response to Reviewer TZXh (Part 4)**
>
> > Explore generalization beyond highly formal tasks. Although the paper acknowledges limitations with open-ended tasks, consider adding an experiment demonstrating partial applicability to semi-structured generation (beyond JSON parsing), such as constrained story generation or semantic parsing with softer constraints.
>
> We thank the reviewer for this suggestion; these are indeed extremely interesting ideas for future work. The focus of our paper is on verifiable outputs and guarantees with objective metrics. However, the expressivity of ASGs could potentially allow soft constraints if the solver is extended appropriately, and hence can be applied to constrained story generation. We will expand on this discussion in the final draft of the paper to open new and exciting avenues for future work in this field.
>
> ---
>
> > Semantic constraint checking can incur significant computational overhead (e.g., >100 seconds in SGS tasks). The absence of wall-clock execution times in Section 5.4 makes it difficult to fully understand or contextualize this overhead relative to end-to-end performance.
>
> To contextualize the computational time of SGS, >100 sec constitutes the majority of the computational time required for constrained decoding, excluding LLM inference time ($T_{LLM}$). We will add further detail and clarify this point in the paper; this refers to $T_{constraints}$ in our previous Formula (1). We also note that this computational time depends on the depth of the sequences’ parse trees in the grammar, which, for this task, is stressed to be higher than standard tasks (32 vs <10 in prior work, e.g., [Allen-Zhu, Z., et al (2024), Delétang, G., et al (2023)]). For other tasks, we note that this time is in the order of 10s of seconds.
>
> ---
>
> ### References
> - Law, M., et al, AAAI, 2019, https://ojs.aaai.org/index.php/AAAI/article/view/4147.
> - Allen-Zhu, Z., et al, 2024, https://arxiv.org/abs/2305.13673.
> - Deletang, G., et al, ICLR, 2023, https://arxiv.org/pdf/2207.02098.
> - Valmeekam, K., et al., TMLR, 2025, https://openreview.net/forum?id=FkKBxp0FhR
> - Zhou, A., et al., ICML, 2024, https://proceedings.mlr.press/v235/zhou24r.html.
> - Wan, Z., et al., ICML, 2024, https://proceedings.mlr.press/v235/wan24c.html.
> - Alviano, M., et al., IJCAI, 2025, https://www.ijcai.org/proceedings/2025/0482.pdf.
> - Ishay, A., et al., KR, 2023, https://proceedings.kr.org/2023/37/kr2023-0037-ishay-et-al.pdf.

---

> > ### Comment · Reviewer_TZXh · 2026-01-29
> > **My concerns have been addressed satisfactorily. Optionally, including examples of the ASGs used in the experiments would be helpful.**
> >
> > I thank the authors for their thorough and thoughtful response. The rebuttal and the updated manuscript address my concerns satisfactorily.
> >
> > In particular:
> > - The argument that the number of tokens serves as a hardware-agnostic metric of computational cost—especially in the context of comparisons with API-based LLMs—is reasonable and convincing. The additional references to prior work that adopt token count as a measure of computation further substantiate this choice.
> > - Regarding the potential barrier to adopting SEM-CTRL due to the need for familiarity with ASPs, the suggestion that LLMs can be used to help generate ASP constraints appears to be a realistic way to mitigate this barrier, given that the generation quality is usable.
> > - I also appreciate the inclusion and clarification of the clock wall time breakdown (formula 1).
> > - Finally, the discussion of ASG limitations—such as the lack of support for fuzzy logic and the exponential worst-case complexity of constraint checking—makes the paper more transparent.
> >
> > As an optional suggestion, it may benefit readers who are not familiar with ASGs to include concrete examples of the constraints used in the experiments, perhaps in an appendix. For example, you mention the ASGs generated for the JSON parsing experiments in Section 5.5, but the manuscript does not appear to include the ASGs themselves. That said, this suggestion is purely optional, and my overall assessment does not depend on whether this point is incorporated.

---

> > > ### Author Response · Authors · 2026-02-12
> > > **Response to Reviewer TZXh (Follow-up)**
> > >
> > > We sincerely thank the reviewer for the careful and thorough evaluation of our response and revised manuscript. We are delighted that our revisions have addressed your concerns.
> > >
> > > Regarding your suggestion to include concrete examples of ASGs from our experiments, we agree that it would indeed benefit readers unfamiliar with ASGs. We have now added Appendix D (Answer Set Grammar Examples, Pages 23-25) to the revised manuscript. This includes four ASG specifications, one from each task class, namely: (1) $a^nb^nc^n$ (Synthetic Grammar Synthesis), (2) Sudoku (Combinatorial Reasoning), (3) Blocksworld, and (4) JSON.
> > >
> > > Thank you again for your suggestions, which have improved the clarity and completeness of our work.

---

### Review · Reviewer_g2Ju · 2026-01-18

**Summary Of Contributions:**

SEM-CRTL is a method for constrained decoding to enforce semantic and syntactic rules on language model generation. The method uses Answer Set Grammars to define semantically valid trajectories that respect the rules of the task, in combination with token-level Monte Carlo Tree Search to search and explore semantically valid trajectories that reach the goal terminating state.

To address concerns of computation efficiency, the implementation relies on caching partial ASG derivations, pruning invalid tokens to reduce the branching factor, and caching partial rollouts of the tree structure.

The authors present a comprehensive set of experiments across a suite of tasks, comparing the performance of off the shelf language models of varying sizes with/without SEM-CTRL to other API accessible models.

Strengths:
* Well-written paper, easy to follow.
* Nice background that makes the paper approachable to a reader outside this domain.
* SEM-CTRL enforces global, context-sensitive constraints.
* Strong empirical performance of SEM-CTRL across various tasks that improves parameter efficiency. SEM-CTRL on a 1b model can achieve higher accuracy than a 70b model.
* SEM-CTRL offers token efficiency compared to reasoning models by constraining the generation.
* Experiments are conducted over a wide range of tasks, validating that SEM-CTRL can be applied broadly.

Weaknesses:
* Introduces computational overhead to check constraints.
* Requires domain knowledge of the user to define the rules and grammar.
* Requires that constraints are binary, which could be limiting in some softer cases.
* Depends somewhat on the alignment of LLM vocab and task vocab.

**Audience:**

Yes

**Audience Explanation:**

Yes, the findings presented in the paper would be of interest to researchers working on applying LLMs to solve tasks with logic-based constraints.

**Broader Impact Concerns:**

No concerns. The authors present the limitations of the paper in a Broader Impact Statement, and there are no ethical implications of the work that need to be addressed.

**Claims And Evidence:**

Yes

**Claims Explanation:**

The authors include substantial empirical evidence that backs the claims they make.

**Requested Changes:**

Minor fixes:
* Page 1: define CSGs before using acronym

---

> ### Author Response · Authors · 2026-01-19
> **Response to Reviewer g2Ju**
>
> We sincerely thank the reviewer for the very positive and encouraging feedback on our work. We are grateful for the recognition of the clarity of presentation, the breadth of experiments, the generality of our work, and the token- and parameter-efficiency gains achieved by $\texttt{SEM-CTRL}$  compared to reasoning-specific and larger models.
> We have carefully considered your feedback, addressed your requested changes, and responded to the weaknesses below:
>
> > Minor fixes: Page 1: define CSGs before using acronym
>
> We will ensure that Context-Sensitive Grammars (CSGs) are explicitly defined upon their first mention in the Introduction of the revised manuscript.
>
> > Introduces computational overhead to check constraints.
>
> As detailed in our response to Reviewer TZXh (Part 2 and 4), we have added a formal decomposition of the wall-clock time to the paper. While constraint checking adds overhead, we note that this is minimal for most tasks (in the order of tens of seconds).
>
> > Requires domain knowledge of the user to define the rules and grammar.
>
> We agree that authoring ASGs requires specific expertise. However, as noted in our response to Reviewer TZXh (Part 3), the syntactic component of ASGs relies on standard CFGs (with specifications and tooling widely available), and recent work suggests that LLMs can assist in translating natural-language constraints into the ASP logic (e.g., [Alviano, M., et al (2025), Ishay, A., et al (2023)]) required by the semantic constraints, thus, reducing the barrier to entry and automating this workflow. We will update our manuscript to make this more explicit, including an overview of how to author new ASGs even in the absence of ASP expertise.
>
> > Requires that constraints are binary, which could be limiting in some softer cases.
>
> Indeed, the current implementation treats constraints as binary (valid/invalid). Extending $\texttt{SEM-CTRL}$ to handle `soft’ constraints is an extremely exciting direction for future work. This can potentially be done by extending the ASG’s solver to handle soft or probabilistic constraints. We will highlight this in the updated manuscript to open exciting new directions for future work.
>
> > Depends somewhat on the alignment of LLM vocab and task vocab.
>
> We agree that misalignment between the LLM’s tokenizer and the grammar’s terminals can affect search efficiency (e.g., by increasing the number of tokens needed to find the optimal completion). However, we note that $\texttt{SEM-CTRL}$ maintains correctness regardless of alignment by masking all invalid tokens at every step, effectively navigating through any tokenization artefacts to ensure the output conforms to the grammar. Given that the token budget and search depth hyperparameters are set sufficiently, $\texttt{SEM-CTRL}$  will find the optimal answer, unlike other approaches, where this may be pruned due to large branching factors without any constraints (e.g., [Wan, Z., et al (2024), Zhang, S., et al (2023)).
>
>
> ### References
> - Allen-Zhu, Z., et al, 2024, https://arxiv.org/abs/2305.13673.
> - Deletang, G., et al, ICLR, 2023, https://arxiv.org/pdf/2207.02098.
> - Alviano, M., et al., IJCAI, 2025, https://www.ijcai.org/proceedings/2025/0482.pdf.
> - Ishay, A., et al., KR, 2023, https://proceedings.kr.org/2023/37/kr2023-0037-ishay-et-al.pdf.
> - Wan, Z., et al., ICML, 2024, https://proceedings.mlr.press/v235/wan24c.html.
> - Zhang, S., et al. ICLR, 2023, https://openreview.net/forum?id=Lr8cOOtYbfL.

---

### Review · Reviewer_3cgA · 2026-01-27

**Summary Of Contributions:**

SEM-CTRL is a decoding framework that guarantees both syntactic and semantic validity of LLM outputs by combining Answer Set Grammars and token-level MCTS. The key insight is that semantic constraints dramatically prune the search space, enabling efficient exploration while guaranteeing validity by construction.

## Key Strengths

- A 1B-parameter model with SEM-CTRL achieves 100% accuracy on tasks where even o4-mini and o1-preview score 75–95%, demonstrating that explicit constraint enforcement outperforms implicit reasoning.
- Unlike prompting or fine-tuning, outputs are provably valid with respect to the specified grammar, so no post-hoc verification needed.
- Uses an order of magnitude fewer tokens than reasoning models while achieving equal or better performance.

## Key Weaknesses

- Authoring ASGs requires a lot of expertise in both the domain and Answer Set Programming, which is a significant adoption barrier.
- This method only works for tasks with well-defined semantics and computable correctness signals; unsuitable for open-ended generation (e.g., summarization).
- ASP constraint checking adds non-trivial latency (up to ~120 seconds on some tasks).
- Benchmarks are pretty narrow and structured; generalization to messier real-world domains is unclear.

**Audience:**

Yes

**Audience Explanation:**

I believe researchers building pipelines requiring guaranteed valid outputs (APIs, code generation, agentic workflows) will be interested in this submission.

**Claims And Evidence:**

Yes

**Claims Explanation:**

I think the claims made in this papre are mostly supported by evidence.

The key claims include:
- semantic constraints (CSEM) contribute more than syntactic constraints (CCFG), and that combining constraints with MCTS yields synergistic gains. (this is supported by ablation studies)
- The 100% validity guarantee (it follows from the construction).
- Token efficiency (the comparisons are pretty straightforward and convincing).

**Requested Changes:**

Overall, I think this paper is in good shape.

---

> ### Author Response · Authors · 2026-01-30
> **Response to Reviewer 3cgA**
>
> We sincerely thank the reviewer for the exceptionally positive and encouraging review of our work. We are grateful for your recognition of SEM-CTRL's key contributions: (1) enabling smaller models (1B parameters) to outperform state-of-the-art reasoning models through explicit constraint enforcement as opposed to implicit reasoning, (2) providing provable validity guarantees that eliminate the need for post-hoc verification, and (3) reducing token consumption by orders of magnitudes while maintaining equal or superior performance. We particularly appreciate your acknowledgment that our claims are well-supported by the ablation studies and empirical evidence presented.
>
> Regarding the weaknesses you identified, we have substantially addressed these concerns in our revised manuscript in response to other reviewers (see our ‘Updated Manuscript’ comment for a summary of changes):
>
> > Authoring ASGs requires a lot of expertise in both the domain and Answer Set Programming, which is a significant adoption barrier.
>
> We have expanded Section 2 (Page 4) and Broader Impact Statement (Page 14) to discuss the authoring process in detail (see also our response to Reviewers TZXh Part 3 and g2Ju). While ASP expertise is beneficial for ASG construction, we note that: (1) the syntactic component (CFG) is widely accessible with existing tooling, and (2) recent work demonstrates that LLMs can assist in translating natural-language constraints into ASP code (e.g., [Alviano, M., et al (2025), Ishay, A., et al (2023)]), lowering the adoption barrier and automating this process. We have incorporated this discussion to make the practical pathway clearer.
>
> > This method only works for tasks with well-defined semantics and computable correctness signals; unsuitable for open-ended generation (e.g., summarization).
>
> We have updated Section 2 (Page 3) to discuss ASG's potential extensibility to soft constraints (see also our response to Reviewer TZXh, Parts 3 and 4). While our current focus is on tasks with verifiable correctness, we note that ASGs could potentially support softer constraints if the underlying ASP solver is extended to handle soft, noisy, or probabilistic constraints. This is an exciting direction for future work, which we highlight in the updated manuscript to encourage future work in this field. We have also extended the Broader Impact Statement (Page 14) to discuss the interpretability advantages of symbolic constraints and their auditability for verification.
>
> > ASP constraint checking adds non-trivial latency (up to ~120 seconds on some tasks).
>
> We have added a formal decomposition of SEM-CTRL's wall-clock time in Section 5.4 (Page 12), clarifying that: (1) the ~120s overhead on synthetic grammar tasks is due to exceptionally deep parse trees (depth 64), explicitly designed to be stressed higher than standard tasks (cf. <10 in prior work, e.g., [Allen-Zhu, Z., et al (2024), Delétang, G., et al (2023)]), and (2) for other tasks, we note that this time is in the order of 10s of seconds (see our response to Reviewers TZXh Parts 2 and 4, and g2Ju for detailed analysis).
>
> > Benchmarks are pretty narrow and structured; generalization to messier real-world domains is unclear.
>
> We agree that our benchmarks focus on structured domains. We selected these tasks precisely because they allow for objective evaluation of both validity and correctness guarantees, which are SEM-CTRL's core contributions. As noted above, we have discussed in Section 2 how ASGs could be extended to support generation with softer constraints for various NLP tasks as future work (i.e., story generation in response to Reviewer TZXh, Part 4), while demonstrating their current applicability to semi-structured domains such as JSON parsing (Section 5.5, Page 12) as a step toward less formal, messier-structured generation.

---

### Author Response · Authors · 2026-01-19
**Updated Manuscript**

We thank all the reviewers and action editor for their comments and suggestions for change. We have now addressed all of them in our revised draft, which we have now uploaded to the system. The major changes are:

- Defined the acronym for Context-Sensitive Grammars (CSGs) in the Introduction (Page 1).

- Updated the Introduction (Page 2) to explicitly distinguish between validity and correctness, and clarified the concept of `global correctness optimization’ with concrete examples.

- Updated the ASG Background section (Page 3) to discuss: (1) the computational power and expressiveness of ASP and types of logic encodable in ASGs, (2) limitations (e.g., fuzzy logic) and potential extensions for soft constraints, and (3) theoretical and computational complexity of ASGs with references to formal proofs.

- Extended the ASG Background section (Page 4) with guidance on the authoring process for new ASGs, including discussion of CFG accessibility, ASP expertise requirements, and LLM-assisted constraint generation to lower adoption barriers.

- Added clarification in the Experimental Setup (Page 9), clarifying that our baselines include methods that perform global correctness optimization.

- Expanded Section 5.4 (Page 12) with: (1) formal decomposition of $\texttt{SEM-CTRL}$'s computational cost, (2) justification for reporting tokens and constraint time as hardware-independent metrics, and (3) contextualization of constraint checking overhead for tasks with deep parse trees.

- Added Section 5.6 (Page 13) comparing fine-tuning with $\texttt{SEM-CTRL}$ and showcasing its complementarity with our method.

- Extended the Broader Impact Statement (Page 14).

We highlight all changes in red in the updated draft.

---

### Decision · Action_Editor_ozkX · 2026-02-20

**Recommendation:** Accept as is

**Audience:**

Yes

**Audience Explanation:**

Researchers and practitioners working on Large Language Models (LLMs), controllable generation, neuro-symbolic methods, structured prediction, and logical reasoning will find this work highly relevant. The ability to guarantee both syntactic and semantic correctness during LLM decoding without the need for fine-tuning or post-hoc verification, addresses major problems in real-world LLM deployments. These include API generation, planning, code generation, and agentic workflows.

**Claims And Evidence:**

Yes

**Claims Explanation:**

All four reviewers unanimously agreed that the claims are well-supported by accurate and convincing evidence. The authors conducted comprehensive empirical evaluations demonstrating that the SEM-CTRL framework allows smaller models (e.g., 1B parameters) to achieve high accuracy and 100% validity on complex tasks, matching or outperforming larger, state-of-the-art reasoning models (e.g., o1-preview, DeepSeek-R1) via explicit constraint enforcement. During the rebuttal phase, the authors further solidified their claims by providing a formal decomposition of computational costs, adding ablation experiments comparing SEM-CTRL w.r.t fine-tuned models, and clarifying the theoretical complexity of Answer Set Grammars (ASGs). The reviewers confirmed that these additions satisfactorily addressed their initial concerns regarding computational overhead, baseline comparisons, and scalability.